# Automatic landmarking identifies new loci associated with face morphology and implicates Neanderthal introgression in human nasal shape

Qing Li [1,24], Jieyi Chen[1,2,24], Pierre Faux [3], Miguel Eduardo Delgado[1,4,5], Betty Bonfante[3], Macarena Fuentes-Guajardo[6], Javier Mendoza-Revilla[7,8], J. Camilo Chacón-Duque [9], Malena Hurtado[7], Valeria Villegas[7], Vanessa Granja[7], Claudia Jaramillo[10], William Arias [10], Rodrigo Barquera [11,12], Paola Everardo-Martínez[11], Mirsha Sánchez-Quinto[13], Jorge Gómez-Valdés [11], Hugo Villamil-Ramírez[14], Caio C. Silva de Cerqueira[15], Tábita Hünemeier[16], Virginia Ramallo[17,18], Sijie Wu[1,2], Siyuan Du [2], Andrea Giardina [19], Soumya Subhra Paria[19], Mahfuzur Rahman Khokan[19], Rolando Gonzalez-José[18], Lavinia Schüler-Faccini[17], Maria-Cátira Bortolini[17], Victor Acuña-Alonzo[11], Samuel Canizales-Quinteros[14], Carla Gallo[7], Giovanni Poletti[7], Winston Rojas[10], Francisco Rothhammer[20], Nicolas Navarro [21,22], Sijia Wang [1,2], Kaustubh Adhikari [19,23,25 ✉] & Andrés Ruiz-Linares [1,3,23,25 ✉]

We report a genome-wide association study of facial features in >6000 Latin Americans based on automatic landmarking of 2D portraits and testing for association with inter-landmark distances. We detected significant associations (P-value $<5 \times 10^{-8}$) at 42 genome regions, nine of which have been previously reported. In follow-up analyses, 26 of the 33 novel regions replicate in East Asians, Europeans, or Africans, and one mouse homologous region influences craniofacial morphology in mice. The novel region in 1q32.3 shows intro-gression from Neanderthals and we find that the introgressed tract increases nasal height (consistent with the differentiation between Neanderthals and modern humans). Novel regions include candidate genes and genome regulatory elements previously implicated in craniofacial development, and show preferential transcription in cranial neural crest cells. The automated approach used here should simplify the collection of large study samples from across the world, facilitating a cosmopolitan characterization of the genetics of facial features.

A full list of author affiliations appears at the end of the paper.

Genome-wide association studies (GWAS) of human facial features are contributing importantly to elucidating the genetic basis of variation in facial features in the general population[1-17]. The genomic regions identified often overlap developmental genes, and have been shown to be enriched in regulatory elements active during craniofacial development[4,17]. These studies were initially performed in individuals of European descent[7,9-11]. Face GWASs are being gradually extended to the characterization populations of non-European ancestry[1-3,5,6,16]. The increasing characterization of non-Europeans is helping to draw a fuller picture of the genetic architecture of facial variation in humans and further our understanding of the evolution of human facial features.

GWASs of facial variation have used a range of phenotyping approaches, from qualitative assessment of morphological features on 2D photographs[1], to measurements based on manual landmarking of 2D photographs[2], to semi-automatic analyses of 3D facial images[4,15]. These approaches vary greatly in cost, informativity, and ease of application. Although 3D images fully represent facial morphology, acquisition of such data requires specialized equipment, complicating their widespread application across the world. Although less informative than 3D imaging, standard 2D photographs have the potential to facilitate the collection of large, diverse study samples. However, manual landmarking of 2D photographs is a slow, labor-intensive task. This has fostered an interest in the application of fully automatic landmarking approaches. However, so far these have enjoyed limited success[18-20]. Most studies based on 2D photographs have therefore been based on entirely manual[1] or, at times, semi-automatic landmarking[9] (i.e. combining automatic landmarking with manual editing).

Here we report a GWAS of facial features derived from a fully automatic landmarking of 2D frontal photographs from Latin Americans of mixed European, Native American and African ancestry. The association signals detected overlap with previous GWAS findings. In addition, we identify 33 novel signals. For most of the novel signals identified, we find evidence of statistical replication in European, East Asian, or African GWAS data, and one mouse homologous region influences craniofacial morphology in mice. One of the novel regions identified includes a tract of introgression from Neanderthals, which we associate with an increase in nasal height, consistent with the morphological differentiation between Neanderthals and modern humans.

## Results

**Study sample and phenotyping.** The 6486 individuals examined here are part of the CANDELA cohort, collected in five Latin American countries[21]. This cohort has been previously studied in GWASs of various physical appearance traits[1,2,22-24]. This includes two previous facial morphology GWASs based on 2D photographs: one mainly based on categorical (i.e. morphoscopic) phenotyping, and one based on manual landmarking of lateral (profile) photographs[1,2,24]. Individuals included in these studies were genotyped on Illumina's OmniExpress chip (including >700,000 SNPs) and characterized for a set of standard covariates (age, sex, BMI, and genetic ancestry estimated from the chip data)[1,2,24].

Here we used the Face++ cloud service platform (https://www.faceplusplus.com) to automatically place 106 landmarks on frontal 2D photographs (i.e. portraits) from the CANDELA individuals (Supplementary Fig. 1). Previously, 16 of these landmarks had been placed manually on a small subset of these individuals[1] and we used these data to evaluate the robustness of the Face++ landmarking. We also compared Face++ with Dlib, a popular landmarking tool[25,26] (Supplementary Table 1). We

calculated Interclass Correlation Coefficients (ICCs) and median Euclidean distances between landmarks placed manually, by Face++, or by Dlib. According to both metrics, the landmarks placed by Face++ were very close to the manual landmarks, and the performance of Face++ was superior to Dlib for certain landmarks (Supplementary Table 1).

After Procrustes superposition, we calculated inter-landmark distances (ILDs) between 34 Face++ landmarks (mostly corresponding to well-defined anatomical landmarks, Fig. 1a and Supplementary Table 2)[1,8,10,17,27-29]. Accounting for face symmetry, we obtained 301 distances. Some of the landmarks retained are on the eyebrow edges, making distances based on them sensitive to eyebrow size (Fig. 1b). The distances obtained show considerable variation and are approximately normally distributed (Supplementary Fig. 2). Many distances show a significant correlation with three head angles estimated by Face++ (pitch, roll, and yaw angle), reflecting the effect of head pose (Supplementary Table 3, Supplementary Fig. 3). Consequently, we excluded 76 individuals with extreme head angle values, and included these angles as covariates in the genetic association tests.

**Trait/covariate correlation and heritability.** A low to moderate (but significant) correlation was detected for various ILDs with covariates (full results are presented in Supplementary Table 3). Strongest correlation with sex was seen for the distances between landmarks 6-4 ($r_{pb} = 0.62$, $p < 10^{-5}$), landmarks 6-25 ($r_{pb} = 0.59$, $p < 10^{-5}$), and landmarks 8-12 ($r_{pb} = 0.58$, $p < 10^{-5}$) (Supplementary Table 3, Supplementary Fig. 3). These three distances are greater in women than in men and relate to eyebrow shape (probably reflecting cosmetic shaping in women). Strongest correlation with age was seen for the distance between landmarks 4-21, sensitive to the spacing between eye and eye-brow ($\rho = -0.31$, $p < 10^{-5}$), and for measures of lip thickness ($\rho = -0.25$, $p < 10^{-5}$, Supplementary Table 3; consistent with previous analyses[1,2]). Strongest correlation with European ancestry was seen for distances sensitive to nasion position (distance between landmarks 12 and 14: $\rho = -0.24$, $p < 10^{-5}$), lip thickness (distance between landmarks 31 and 33: $\rho = -0.19$, $p < 10^{-5}$), nasal root breadth (distance between landmarks 14 and 15: $\rho = -0.22$, $p < 10^{-5}$) and nose wing breadth (distance between landmarks 16 and 17: $\rho = -0.20$, $p < 10^{-5}$) (Supplementary Table 3, Supplementary Fig. 3). These correlations of facial features with genetic ancestry are consistent with previous observations[1,2]. We estimated narrow-sense heritability ($h^2$) based on the kinship matrix derived from SNP data[30] and observe moderate values for most traits (Median $h^2$ of 0.38, Supplementary Fig. 3 and Supplementary Table 3).

**Overview of GWAS results.** After applying genotype and phenotype data quality control (QC) filters (see Methods for details), we evaluated association for ILDs with up to 11,532,785 SNPs on up to 5988 individuals. We considered a $P$-value $< 5 \times 10^{-8}$ as threshold for significance, as this is stricter than the False Discovery Rate (FDR) multiple testing correction procedure of Benjamini–Hochberg (which results in a threshold of $2 \times 10^{-6}$, across SNPs and traits, see Methods). Altogether, 42 genomic regions were significantly associated with at least one ILD and 148 distances were associated with at least one of these 42 genomic regions (Fig. 1c, Supplementary Table 4). Among these 42 regions, nine have been previously reported in previous GWAS of facial features, including six regions that were detected in the two previous face GWASs we conducted in the CANDELA cohort (Supplementary Fig. 4 and Supplementary Table 4). Table 1 provides summary information on the nine regions reported in previous studies that were replicated here (additional

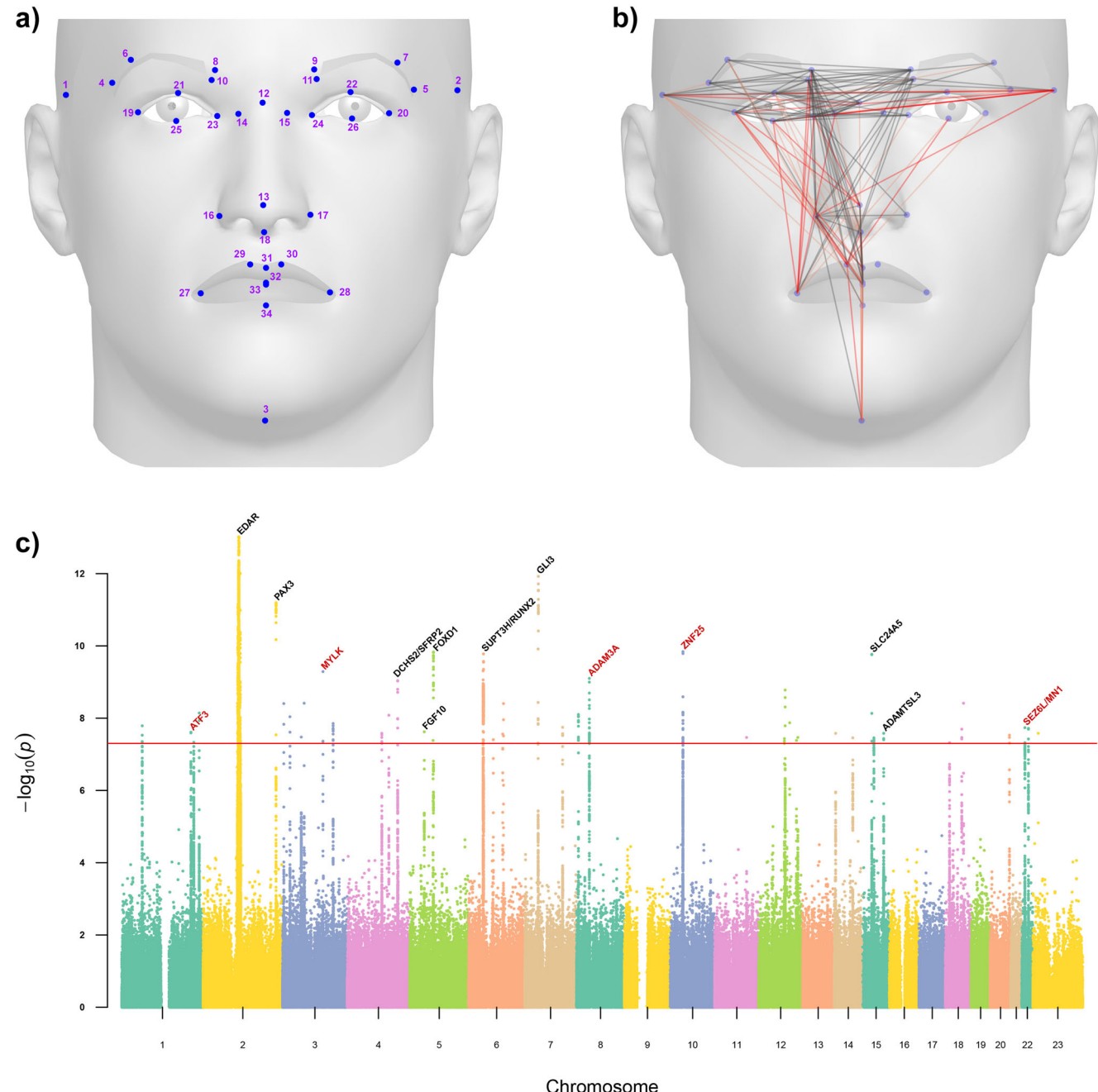

**Fig. 1 Overview of the facial features GWAS performed here. a** Dots indicate the location of the 34 facial landmarks used for calculation of 301 inter-landmark distances (ILDs, Supplementary Table 2 provides additional information on these landmarks). **b** Lines represent the 148 ILDs for which we detected significant association with at least one genomic region in the CANDELA data. Black lines refer to ILDs that resulted in replication of previously reported associations (Table 1). Red lines represent ILDs that revealed novel associations (with a darker red highlighting the ILDs associated with five genomic regions discussed in the text; Table 2). **c** Combined Manhattan plot illustrating all significant GWAS hits (-log(P) > 7.3, red line). Black labels indicate candidate genes from previous GWASs that were replicated here (Table 1). Red labels highlight the main candidate genes at the five novel regions discussed in the text. For visibility, the y-axis has been truncated at a -log(P) of 13.

information on these regions is provided in Supplementary Note 1 and Supplementary Fig. 5).

**Follow-up of newly associated regions: replication in independent cohorts.** We sought evidence of replication for the 33 newly associated genome regions using results from studies in independent samples. Considering the admixed ancestry of the CANDELA individuals, we sought replication in samples with different continental ancestries. We therefore used available data

from East Asians, Europeans and Africans. For East Asians, we had available frontal 2D photographs and genome-wide SNP data for 5078 individuals[31,32]. These data were processed as for the CANDELA sample. For Europeans and Africans, we extracted association P-values from published studies: a GWAS meta-analysis including data for 10,115 Europeans and 78 ILDs[17] and a GWAS performed in 3631 Africans for 34 size and shape-related facial traits (distances and Principal Components (PCs))[5,33]. When data for the index SNP of a region identified in the CANDELA sample was not available in the other study samples,

| Chromosomal region | Index SNP | Candidate gene | # significant ILDs | # significant SNPs | Strongest P-value | Refs. |
|---|---|---|---|---|---|---|
| 2q12.3 | rs72627476 | **_EDAR_** | 75 | 1245 | $5.60 \times 10^{-34}$ | 1,2,8 |
| 7p14.1 | rs846315 | _GLI3_ | 5 | 31 | $1.20 \times 10^{-12}$ | 1,15 |
| 2q36.1 | rs13022712 | _PAX3_ | 18 | 37 | $6.42 \times 10^{-12}$ | 2,4,8–11,15,17 |
| 5q13.2 | rs7341037 | FOXD1 | 10 | 18 | $1.49 \times 10^{-10}$ | 31 |
| 6p21.1 | rs141680515 | SUPT3H/RUNX2 | 15 | 275 | $1.68 \times 10^{-10}$ | 1,2,4,8,11,15 |
| 15q21.1 | rs1426654 | **_SLC24A5_** | 2 | 4 | $1.74 \times 10^{-10}$ | 2 |
| 4q31.3 | rs2045323 | DCHS2/SFRP2 | 8 | 64 | $9.35 \times 10^{-10}$ | 1,2,4,15 |
| 5p12 | rs4505960 | FGF10 | 3 | 1 | $2.43 \times 10^{-8}$ | 15 |
| 15q25.2 | rs62027787 | _ADAMTSL3_ | 1 | 3 | $2.65 \times 10^{-8}$ | 15 |

**Table 1 Features of nine genome regions reported in previous face GWASs for which genome-wide significant association is also observed here.**

Genes underlined include significantly associated SNPs. Genes in bold include an associated SNP leading to an amino-acid substitution.

we examined as proxies SNPs in LD with the index SNP in a region ($r^2 >= 0.1$). For six of the novel regions detected, no polymorphic SNPs across datasets were available, preventing evaluation of replication for these regions. We calculated a significance threshold for replication of 0.029 using Benjamini–Hochberg's FDR procedure (accounting for 27 regions tested in four replication datasets). Altogether, 26/33 regions had association P-values <0.029 for at least one distance, in at least one of the replication datasets (22 in East Asians, 21 in Europeans and 5 in Africans; with 4 regions replicating in all three independent datasets) (Supplementary Table 4, Supplementary Note 2-3).

**Follow-up of novel face regions in the mouse**. To evaluate the potential effect in the mouse of the face regions newly identified here, we reanalyzed published genome-wide SNP data from outbred mice characterized for craniofacial shape variation[34]. Of the 33 novel regions identified here, 30 could be successfully mapped onto the mouse genome (Supplementary Table 5). Of these, a region on mouse chromosome 5q (homologous to human 22q12.1) showed significant association for SNPs over a ~1.5 Mb segment, with the index SNP in this region (rs32069343, P-value: $2 \times 10^{-34}$), impacting on multiple aspects of mouse skull and mandible shape (Fig. 2, Supplementary Movie). In the CANDELA GWAS, SNPs in 22q12.1 are associated with ILD D437 between landmarks 3 and 31 (Fig. 1), a distance sensitive to the height of the lower face (smallest P-value of $1.8 \times 10^{-8}$ for rs9608473, Fig. 2). In previous studies, SNPs in 22q12.1 have been strongly associated with height[35], and suggestively associated with facial features[36,37] and cleft lip/palate[38].

**Neanderthal introgression in 1q32.3 and facial morphology**. One of the novel, replicating, regions identified here is in 1q32.3. SNPs in this region are associated with ILDs D203, D166 and D233, which involve landmark 13 together with landmarks 23, 19 and 25, respectively (Fig. 1). Strongest association was observed for rs12564392 and ILD D203 (P-value $2 \times 10^{-8}$). The three associated distances are mainly sensitive to midface height. Interestingly, previous studies have reported Neanderthal introgression in 1q32.3[39,40]. To evaluate the relationship between the association signal in 1q32.3 and Neanderthal introgression in the region we screened a 1 Mb window around the association signal for evidence of introgression in the CANDELA data[41]. Considering only introgression tracts >10 Kb long called at >99% confidence, we observe that Neanderthal introgression in 1q32.3 peaks in the region of strongest association seen in the GWAS (Fig. 3). Up to 31% of CANDELA chromosomes carry Neanderthal tracts in this region. As seen in the SNP-based GWAS, the Neanderthal tracts are significantly associated with distances

D203, D166, D223 (at a Benjamini–Hochberg's FDR significance threshold of 0.015), and lead to an increase of these distances (Fig. 3, Supplementary Table 6).

To evaluate the consistency of the introgression effect seen in the CANDELA data with the facial differentiation between modern humans and Neanderthals we examined available data on Neanderthal facial features[42]. No information is available in Neanderthals for distances equivalent to D203, D166 or D223. However, a related distance (also sensitive to midface height) which is available in Neanderthal is Subspinale-Nasion (i.e. nasal height). The equivalent distance, between Subnasal (landmark 18) and Nasion (landmark 12), was also measured in the CANDELA individuals (ILD D117). We thus tested for association between Neanderthal introgression in 1q32.3 and D117 and found it to be significant (P-value $1.7 \times 10^{-7}$; Fig. 3, Supplementary Table 6), with introgression resulting in an increased distance. Consistently, comparison of skulls from modern humans and Neanderthals shows that Neanderthals have a markedly higher nasal height (Fig. 3, Supplementary Table 7).

Local ancestry analyses in the CANDELA individuals show that Neanderthal tracts occur almost exclusively on a Native American chromosomal background (Supplementary Fig. 6). This observation is consistent with previous analyses which detected 1q32.3 introgression essentially in Native Americans[40] and agrees with the GWAS index SNP in this region (rs12564392) having highly differentiated allele frequencies between Europeans and Native Americans (Table 2).

**Features of novel regions and their effects on ILDs**. Table 2 summarizes key features for the 22q12.1 and 1q32.3 regions discussed above as well as for the three novel (replicating) regions, most strongly associated with ILDs in the CANDELA sample. Association plots for these three regions and the associated ILDs are shown in Fig. 4. Similar information on all the other novel associated regions is presented in Supplementary Table 4 and Supplementary Note 2-3. SNPs in 3q21.1 are associated with 8 distances reflecting variation mainly in the width of the upper face with strongest association being seen with distance D213 (between landmarks 2 and 25, Figs. 1 and 4). SNPs in 8p11.21 are associated with seven distances (strongest association seen for rs59547557 with D332, involving landmarks 10 and 27, P-value $2.24 \times 10^{-9}$, Fig. 4). All seven distances associated with this region are sensitive to the position of the right cheilion (Figs. 1 and 4). In previous studies, SNPs in 8p11.21 have been reported to be suggestively associated with non-syndromic cleft lip/palate[43]. SNPs in 10p11.1 are associated with 8 distances, with strongest association seen for rs58831446 with D511 (between landmarks 16 and 33, P-value $1 \times 10^{-10}$). These 8 distances are sensitive to philtrum height (Figs. 1 and 4).

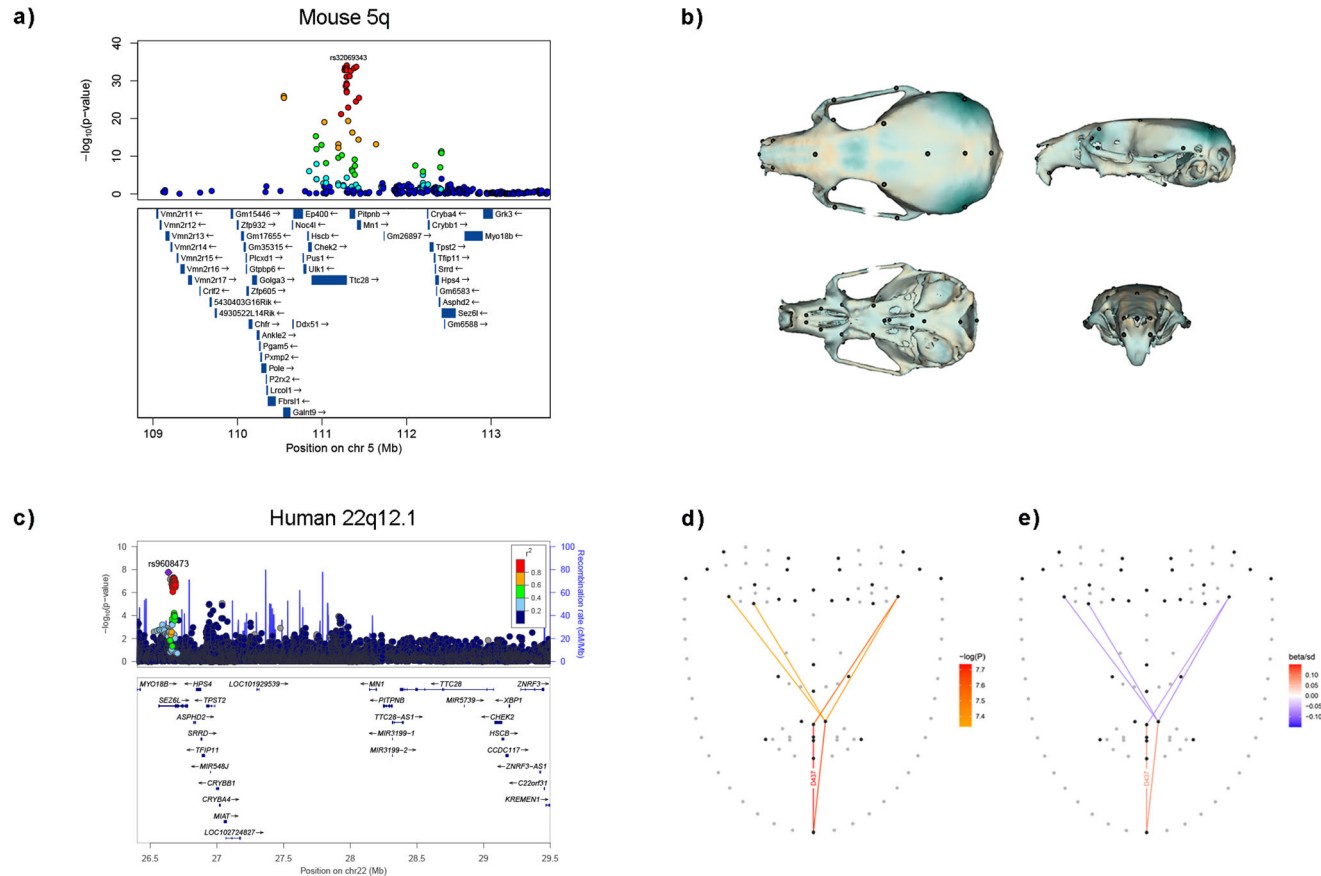

**Fig. 2 Regional association plots in mouse and human. a** At the top are shown *P*-values for SNPs in the mouse chromosome 5 region homologous to human 22q12.1 . Dot colors reflect LD with the index SNP (rs32069343). Underneath are shown genes annotated in the region. **b** Effect of rs32069343 on mouse skull shape (expansion/contraction relative to the mean shape is shown in blue/brown). To facilitate visualization, the effect has been magnified ×20 (see also Supplementary Movie). **c** LocusZoom plot for 22q12.1: the top panel shows SNP *P*-values (dot color reflecting LD with the index SNP rs9608473). The bottom panel displays genes annotated in the region. **d**, **e** Facial landmarks placed by Face++ (dark dots indicate the landmarks retained in ILD calculation). Lines indicate ILDs associated with SNPs in 22q12.1. In **d** line color reflects association *P*-value; in **e** line color reflects the direction and magnitude of the association. The ILD with strongest association P-value (D437) is labelled.

**Genome annotation, Gene Ontology and transcription patterns in associated regions**. We used FUMA[44] to examine genome annotations for the 186 SNPs that were significantly associated across the 33 novel regions detected in the CANDELA sample. Altogether, 91 are intergenic, 55 are intronic, 39 are ncRNA variants, and one is in a 3' untranslated region. In line with previous analyses showing an enrichment of SNPs associated with facial features in regulatory elements active during craniofacial development[4,17], we observe that SNPs in the novel regions identified here are usually near or within known craniofacial enhancers/promoters (e.g. 1q32.3 and 12q21.31, Fig. 3, Supplementary Table 4). We performed a Gene Ontology (GO) analysis for the genes nearest to the index SNPs of the novel associated regions. Consistent with previous analyses[4,17], we found that these genes are significantly enriched in growth and development terms, including: GO:0006936: muscle contraction (*P*-value = $8.81 \times 10^{-5}$), GO:0019827: stem cell population maintenance (*P*-value = $2.19 \times 10^{-4}$), GO:0051960: regulation of nervous system development (*P*-value = $1.17 \times 10^{-3}$), GO:0021700: developmental maturation (*P*-value = $2.28 \times 10^{-3}$), GO:0048562: embryonic organ morphogenesis (*P*-value = $3.30 \times 10^{-3}$), GO:0007162: negative regulation of cell adhesion (*P*-value = $3.83 \times 10^{-3}$), and GO:0032940: secretion by cell (*P*-value = $8.17 \times 10^{-3}$) (Supplementary Fig. 7A). To evaluate preferential transcription of genes in the newly associated regions, we contrasted publicly available RNAseq data from cranial neural crest cells (CNCC)[45] to data for 318 other cell types obtained by the ENCODE project[46]. We found that, for the majority of the regions that could be tested (19/26), transcripts closest to the index SNPs are preferentially expressed in CNCCs, compared to other cell types, similar to what has been observed in previous analyses[4,17] (Supplementary Fig. 7B).

## Discussion

GWAS of facial features have identified dozens of associated genome regions[1–17]. In some cases, these regions overlap genes for which experimental evidence demonstrates their involvement in craniofacial development[1,2,47]. Furthermore, most of the SNPs associated with facial features are in non-coding regions, and enrichment analyses indicate that these SNPs are preferentially located in regulatory elements of the genome, active during craniofacial development[1,2,4,17]. Consistently, the novel loci (and associated SNPs) we identify here share features with findings from previous GWAS of facial morphology.

Considering the five chromosomal regions highlighted in Table 2: (i) 1q32.3 overlaps the Activating Transcription Factor 3 *ATF3* gene (Fig. 3). *ATF3* is an evolutionarily highly conserved transcription factor known to be involved in nervous tissue regeneration after trauma[48]. Although currently there is no evidence for a direct involvement of *ATF3* in craniofacial development, it has been reported that *ATF3* expression is regulated by *FOXL2*, a transcription factor whose mutations are known to lead

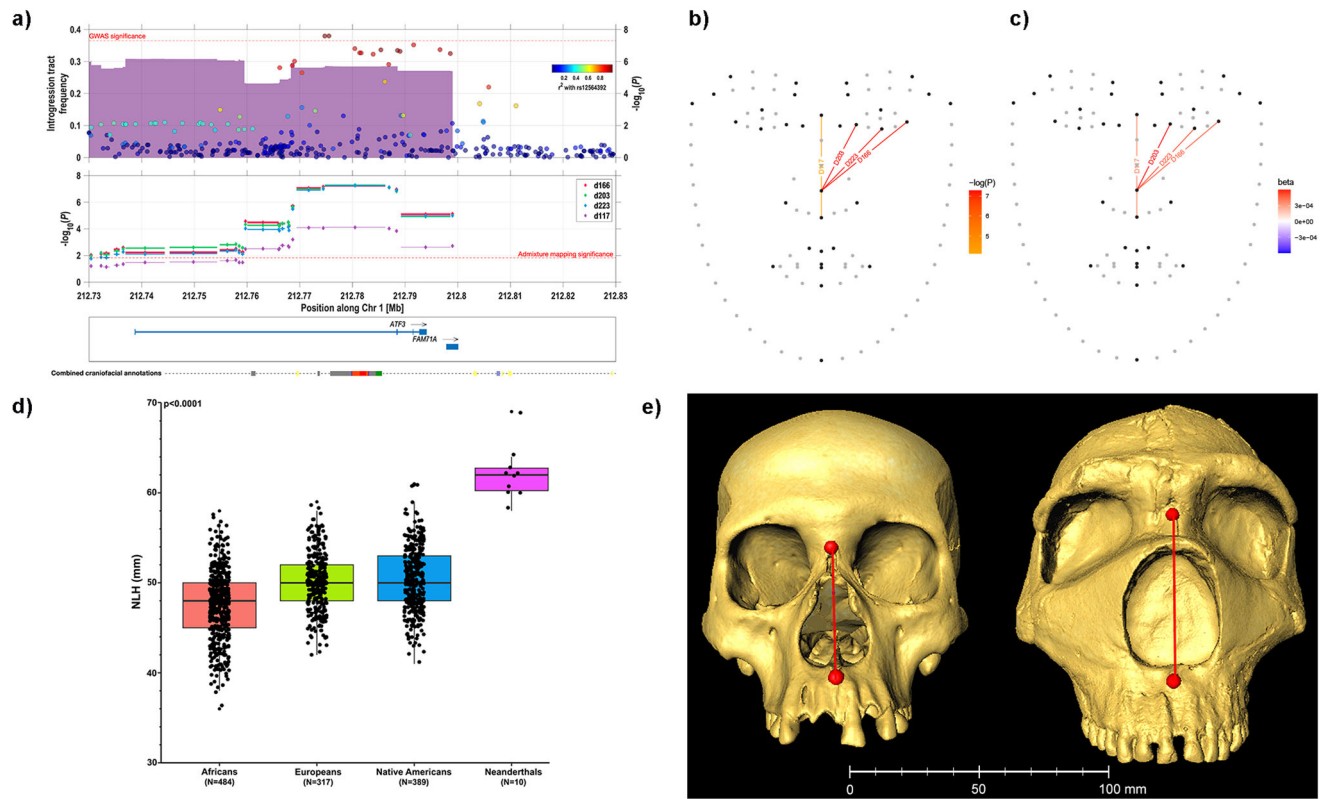

**Fig. 3 Neanderthal introgression in 1q32.3 and facial variation in the CANDELA sample. a** At the top is shown the frequency of introgression tracts (i.e. stacking tracts across CANDELA individuals) overlaid onto the GWAS SNP *P*-values with ILD D223. The 100 kb segment displayed includes the GWAS and introgression peaks. Below are shown association *P*-values from an admixture mapping analysis of the Neanderthal introgression tracts with four ILDs (D166, D203, D223 and D117). Diamonds mark the center of an admixture mapping segment with whiskers marking its extent (details on these segments are in Supplementary Table 6). At the bottom are shown genes in the region. The dotted line beneath the genes displays craniofacial annotations from the Epigenomic and Transcriptomic Atlas of Human Craniofacial Development (colored in accordance with the Roadmap Epigenomics project https://egg2.wustl.edu/roadmap/web_portal/imputed.html#chr_imp; red/brown boxes representing craniofacial-specific super-enhancers). **b**, **c** ILDs tested in panel **a**: D166 (landmarks 13–19), D203 (landmarks 13–23), D223 (landmarks 13–25), and D117 (landmarks 12–18). In **b** the color scale indicates admixture mapping association *P*-value. In **c** color scale indicates direction and magnitude of the association. **d** Box plots for nasal height (NLH) in skulls from 1190 modern humans (from three continental populations) and 10 Neanderthals. Modern human data is from Howells' database (http://web.utk.edu/~auerbach/HOWL.htreilm;[94] Supplementary Table 7). Neanderthal data is from Weaver and Stringer[95]. *P*-value shown is from a Mann–Whitney U test contrasting modern human and Neanderthal data. **e** Example skulls from a modern human (Native American) and a Neanderthal (Amud 1). The 3D images are reproduced on the scale shown underneath. Nasal height (distance between the nasion and sub-spinale landmarks) is shown as a red line (modern human = 50.2 mm; Neanderthal = 63.8 mm). The modern human image is from the collection of the División de Antropología, Museo de La Plata (Argentina). The Neanderthal image was obtained from the MorphoSource repository (https://www.morphosource.org/concern/media/000005749).

to alterations of the midface[49]. Furthermore, strongest association was observed for SNPs intronic to *ATF3* around an enhancer which has been shown to be active during craniofacial development[50] (Fig. 3). (ii) Associated SNPs in 3q21.1 overlap the *MYLK* (Myosin light chain kinase) gene, which studies in mice have implicated in palate fusion during development[51]. (iii) The newly associated 8p11.21 region includes a cluster of disintegrin and metalloproteinase (*ADAM*) domain genes (Fig. 4). This is a family of surface proteins with adhesion and protease activity, members of which have been shown to be involved in craniofacial development[52]. Furthermore, one of the ADAM genes in the cluster on 8p11.21 (*ADAM3A*), has been suggestively associated with non-syndromic cleft lip/palate[43]. (iv) Associated SNP on 10p11.1 overlap a cluster of Zinc Finger proteins genes (*ZNF*, Fig. 4). This cluster includes *ZNF25*, which has been shown to be involved in osteoblast differentiation of human skeletal stem cells[53], this is a process in which *RUNX2* (a well-established craniofacial morphology gene, Table 1) also plays a major role[54–56]. (v) The mouse analyses performed here are consistent with the novel association we detect on human 22q12.1. In

humans, maximum association is seen for SNPs intronic to the *SEZ6L* gene (Fig. 2). In mice, SNPs in *Sez6l* are also significantly associated, although association is strongest around the *Ttc28* and *Mn1* gene regions (Fig. 2). There is currently no evidence implicating *SEZ6L* directly in craniofacial phenotypes, but there is abundant evidence that *Ttc28, Mn1* and other genes in this region are involved in mouse craniofacial development (Fig. 2)[34,57,58]. Interestingly, of the candidates highlighted here, three (*ATF3, MYLK* and *SEZ6L*) are the genes closest to the index SNPs and, in our RNAseq data analysis, we observe that two of these genes (*MYLK* and *SEZ6L*) are preferentially transcribed in CNCC cells (Supplementary Fig. 7B).

Genetic determinants of variation in facial features in contemporary human populations are also likely to have played a role during the evolution of facial morphology. We previously identified a region in 1p12 in which a tract introgressed from archaic humans (Denisovans) impacts on lip thickness. That chromosomal region had previously been shown to be associated with body fat distribution[59] and bears a strong signature of natural selection, raising the possibility that Denisovan introgression could have

**Table 2 Features of the five novel face loci discussed in the text[a].**

| Region | Index SNP | Alleles (Ref/Alt) | Association P-value | # Significant ILDs | # Significant SNPs | Replication P-value[b] | Alternative allele frequency[c] | | | | Main candidate genes |
|---|---|---|---|---|---|---|---|---|---|---|---|
| | | | | | | | CAN | NAM | EUR | AFR | |
| 1q32.3 | rs12564392 | C/A | $2 \times 10^{-8}$ | 3 | 3 | $1 \times 10^{-2}$ | 0.28 | 0.61 | 0.01 | 0.00 | ATF3 |
| 3q21.1 | rs820360 | A/G | $5 \times 10^{-10}$ | 8 | 3 | $1 \times 10^{-3}$ | 0.35 | 0.32 | 0.37 | 0.06 | MYLK |
| 8p11.21 | rs59547557 | T/C | $7 \times 10^{-10}$ | 7 | 39 | $3 \times 10^{-3}$ | 0.48 | 0.31 | 0.55 | 0.81 | ADAM3A |
| 10p11.1 | rs58831446 | A/AT | $1 \times 10^{-10}$ | 8 | 25 | $2 \times 10^{-4}$ | 0.51 | 0.66 | 0.41 | 0.44 | ZNF25 |
| 22q12.1 | rs9608473 | G/A | $1 \times 10^{-8}$ | 6 | 11 | $1 \times 10^{-3}$ | 0.29 | 0.49 | 0.10 | 0.07 | SEZ6L/MN1 |

a Information on all the regions detected is in Supplementary Table 4.
b Smallest P-value observed in the Chinese or European replication samples.
c Alternative allele frequency of index SNP in: CANDELA (CAN); Sub-Saharan Africans (AFR); Europeans (EUR)[97] and Native Americans (NAM)[76].

facilitated adaptation to a cold environment. The evidence we observe of Neanderthal introgression in 1q32.3 impacting on mid-face height represents the second instance of archaic human introgression affecting facial morphology in modern humans. In this case, the possibility of examining similar skull traits in contemporary human and Neanderthal skulls allowed us to determine that the increase in mid-face height associated with archaic introgression in 1q32.3 is consistent with the modern human-Neanderthal morphological differentiation. Evaluating the consistency of phenotypic effects had not been possible in the case of Denisovan introgression in 1p12 as that case concerned only soft tissues (the lips). Analysis of skulls has long shown that facial morphology differs markedly between Neanderthals and modern humans with the mid-face, particularly the nasal cavity, showing major differences[60]. This includes markedly taller noses in Neanderthals than in modern humans. Furthermore, it has long been speculated that nose morphology (in Neanderthals as well as in modern humans) has been the subject of natural selection, particularly as an adaptation to environmental temperature and humidity[61–63]. Further genetic work, including future analyses of additional ancient DNA samples, could help shed light on this question.

Although the earliest (and largest) studies on the genetics of facial variation have been carried out in people of Europeans ancestry[4,9,10], recent efforts have increasingly sought to examine non-Europeans[1,2,5,32]. Populations with admixed continental ancestry, such as Latin Americans, offer challenges and opportunities for such studies. In these populations, optimal correction for population stratification, considering both global and local genomic ancestry, is a challenging analytical problem for which an all-round solution is yet to be developed[64–66]. Use of genetic PCs and local ancestry correction approaches to deal with population stratification should therefore be undertaken with caution. Nevertheless, the extensive genetic and phenotypic diversity of Latin Americans is enabling GWASs that have led to important insights into the genetics of physical appearance[1,2,22–24]. This is illustrated here by the novel instance of archaic introgression detected in 1q32.3: the introgressed tract has a high frequency in Native Americans but is essentially absent in Europeans (Table 2, Supplementary Fig. 6). Given the widespread availability of 2D photographs, the automated landmarking approach we applied here could facilitate a more comprehensive world-wide sampling of human facial variation than hitherto attempted. The study of larger and more diverse study samples should enable a fuller assessment of the genetic architecture of facial variation in the global human population and of the evolutionary forces that have shaped this variation across the world.

## Methods

**Study subjects**. Discovery sample: 6486 (Colombia, $N = 1407$; Brazil, $N = 674$; Chile, $N = 2003$; Mexico, $N = 1203$ and Peru, $N = 1199$) individuals from the Consortium for the Analysis of the Diversity and Evolution of Latin America (CANDELA consortium) were included in frontal photographs collection. CANDELA consortium (https://www.ucl.ac.uk/biosciences/gee/candela/) has been used to study physical appearance in Latin American for multiple studies, and details could be seen in Ruiz-Linares et al.[21]. Ethical approval was obtained from the Universidad Nacional Autónoma de México (México), Universidad de Antioquia (Colombia), Universidad Perúana Cayetano Heredia (Perú), Universidad de Tarapacá (Chile), Universidade Federal do Rio Grande do Sul (Brazil) and University College London (UK). All participants provided written informed consent. The participate included in Supplementary Table 1 has also provided written informed consent and signed Research Participant Release Form.

Replication samples: We examined replication in three independent data samples: one Chinese, one European, and one African cohort (one SNP GWAS, and one CNV GWAS).

The Chinese sample includes 5298 individuals[32]. This sample stems from the National Survey of Physical Traits (NSPT) cohort ($n = 2628$) and the Taizhou Longitudinal Study (TZL) cohort ($n = 2670$)[31,67]. The Taizhou Longitudinal

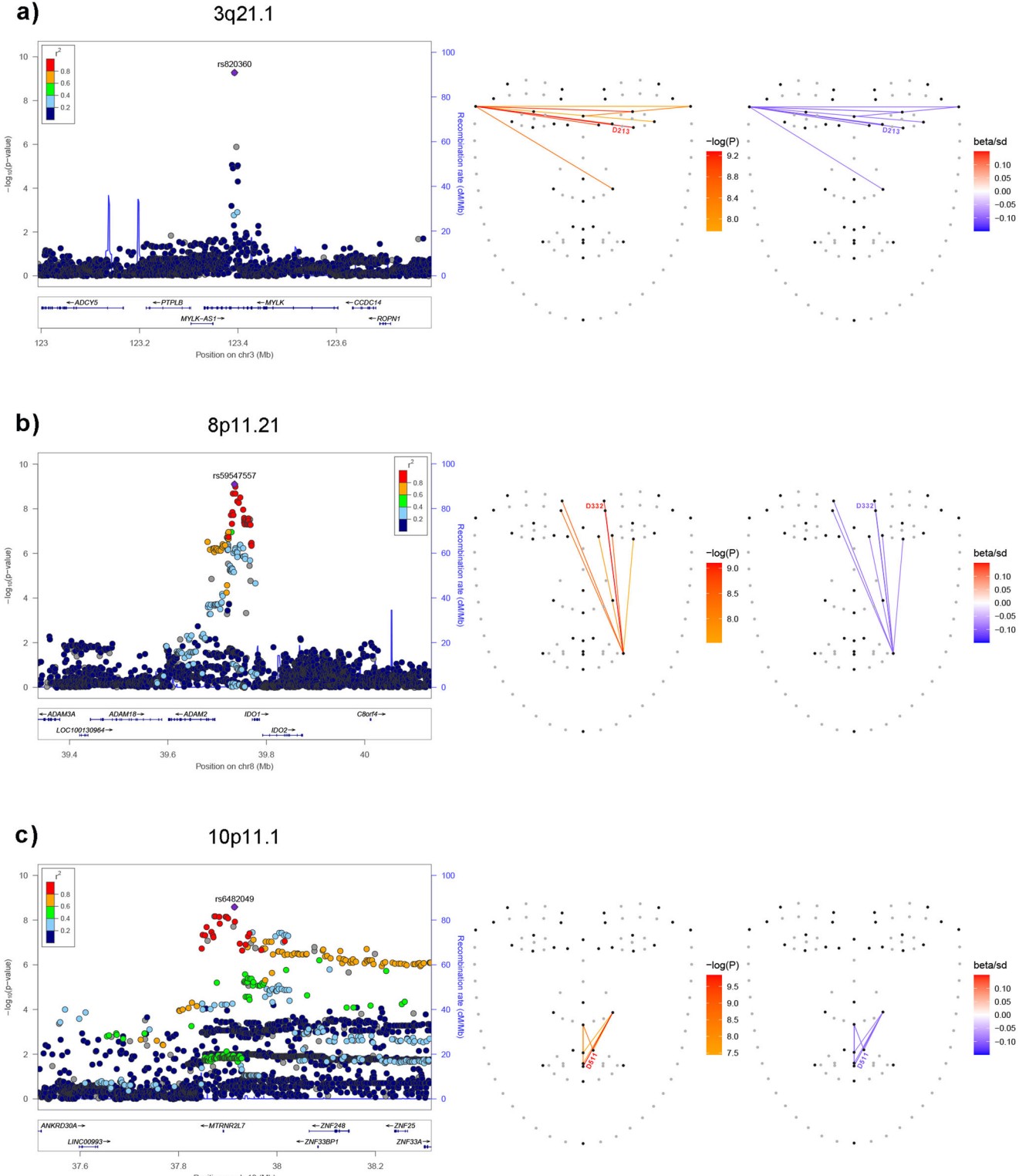

**Fig. 4 Regional association plots for the three novel genomic regions showing strongest association with facial features in the CANDELA sample.**
Panels **a–c** show on the left the association *P*-values for SNPs in each region (index SNP has been labeled). Annotated genes in each region are shown underneath. To the right of each panel are shown the facial landmarks placed by Face++ (dark dots indicate the landmarks retained in ILD calculation). Associated ILDs are indicated with colored lines (left face: line color reflects direction and magnitude of the association; right face: line color reflects association *P*-value). The ILD with strongest association *P*-value is labeled.

Study (TZL) was approved by the Ethics Committee of Human Genetic Resources at the Shanghai Institute of Life Sciences, Chinese Academy of Sciences (ER-SIBS-261410). The National Survey of Physical Traits (NSPT) is the sub-project of The National Science & Technology Basic Research Project which was approved by the Ethics Committee of Human Genetic Resources of

School of Life Sciences, Fudan University, Shanghai (14117). All participants provided written informed consent.

The European replication sample is the discovery cohort examined in the GWAS of Xiong et al.[17]. This sample includes 10,115 individuals of European ancestry recruited in three countries (Netherlands, *N* = 3193; United Kingdom,

$N = 4727$ and United States, $N = 2195$). The summary statistics are publicly available at https://doi.org/10.6084/m9.figshare.10298396[68].

The African cohort is the discovery cohort examined in a CNV GWAS of Null et al.[33], and a SNP GWAS of Cole et al.[5]. This sample contains 3631 Bantu African individuals aged from 3 to 21 from the Mwanza region of Tanzania. The summary statistics are available at https://github.com/meganmichelle/CNV_FaceShape and https://www.ncbi.nlm.nih.gov/projects/gap/cgi-bin/study.cgi?study_id=phs000622.v1.p1.

**Genotype data**. The genotype data examined here are those analyzed in previous GWAS of the CANDELA sample[1,2,22–24]. Briefly, a blood sample was collected from each volunteer and DNA extracted following standard laboratory procedures. DNA samples were genotyped on the Illumina HumanOmniExpress chip including 730,525 SNPs. PLINK v1.90 was used for QC. Individuals and SNPs with >5% missing genotypes, SNPs with <1% minor allele frequency, and individuals who failed the X- or Y- chromosome sex checks were excluded. After these QC filters, ~650,000 SNPs and 5500 individuals were retained for further analyses. Human genome reference assembly GrCh37/hg19 was used. SHAPEIT2[69] was used for pre-phasing the chip genotype data, and IMPUTE2[70] was then used to impute variants using the 1000 Genomes Phase 3 reference panel. Imputation led to 11,532,785 SNPs being available for association testing. Markers that are monomorphic in 1000 Genomes Latin American samples were excluded from imputation. Chip genotyped SNPs having a low concordance value (<0.7) or a large gap between info and concordance values (info_type0 – concord_type0 > 0.1), which might be indicators of poor genotyping, were also removed, both from the imputed and chip dataset. Imputed SNPs with imputation quality scores <0.4 were excluded. The IMPUTE2 genotype probabilities at each locus were converted into best-guess genotypes using PLINK (at the default setting of <0.1 uncertainty). SNPs with >5% uncalled genotypes or minor allele frequency <1% were excluded. On the basis of genome-wide SNP data, we estimated European, Native American and sub-Saharan African ancestry proportions for each CANDELA individual (European and Native American ancestries being strongly negatively correlated[21]).

**Phenotyping**. Frontal digital photographs were taken for each CANDELA volunteer, at eye level, 1.5 m away, using a Nikon D90 camera (12,3 Megapixels resolution) fitted with a Nikkor 50 mm fixed-focal-length lens[21]. The photographs were anonymized for confidentiality, and stored on a secure cluster, where an API script available from Face++ (https://www.faceplusplus.com), implementing a pre-trained deep learning model was run. Face++ placed 106 landmarks on each photograph (Supplementary Fig. 1), and provided a set of attribute values. Face images with attribute values indicative of poor quality (e.g. blurriness, head rotation represented through three head angles) were excluded. Individuals presenting an outlier phenotype value (trait value lower or greater than the trait value average for that sex ±three times the standard deviation) were also removed for each phenotype. We focused on 34 landmarks corresponding mostly to well-defined anatomical landmarks of common usage[27,28] including previous GWAS studies (Fig. 1a, Supplementary Table 2)[1,8,10,17,29]. Specifically, 28 out of 34 landmarks are well-defined anatomical landmarks, while the other six landmarks (these are more commonly referred to as "semi-landmarks" or "pseudo-landmarks" in the physical anthropology literature[2], but to simplify the presentation, are referred collectively as "landmarks" too) are on important locations such as the end of a contour, which would allow us to capture facial features that we are interested in, including face width and eyebrow size (Supplementary Table 2). Procrustes superimposition was performed using MorphoJ[71] and pairwise ILD calculation was carried out using R[72]. Since Procrustes-adjusted landmarks coordinates were symmetrized, some ILDs were identical. After removing 260 duplicates, 301 distances (labeled as 'D' followed by a number, Supplementary Table 3) were retained for the GWAS.

To evaluate the robustness of the Face++ landmarking, we examined the accuracy of 32 landmarks which were either placed manually on a subset of 1610 photographs in a previous study[1], or included in another commonly used automatic face landmarks detection protocol Dlib[25,26] on the same 1610 photographs (Supplementary Table 1). Median Euclidean distances and ICCs between Face++ and manual landmark coordinates, and between Face++ and Dlib landmark coordinates were obtained using Matlab[73]. Generally, the consistency of Face++ landmarks compared to manual landmarks were similar to, and for some landmarks better than Dlib, according to both ICC and pixel distance (Supplementary Table 1). Also, Face++ provided more landmarks (106) than Dlib (68), especially in anatomically important regions such as nasal bridge which have been associated with genomic regions in previous studies[1]. Therefore, we eventually chose Face++ beyond Dlib as our automatic landmarking tool.

**Statistical genetic analysis**. We used point biserial correlation coefficient ($r_{pb}$) to test the correlation of ILDs with gender and Spearman's correlation coefficient ($\rho$) to test the correlation of ILDs with age, BMI, genetic ancestry and head angles.

Relatedness between samples was estimated using KING-robust[74] implemented in PLINK v2.0, which is better suited to estimate relatedness in admixed individuals. Only one individual from any related pair (with a threshold of IBD > 0.1, excluding third degree relatives and higher) was retained. To estimate the narrow-sense heritability ($h^2$) for each trait, we computed a genomic

relationship matrix (GRM) combining genotype data for of all individuals for which data for at least one trait was available. The GRM was calculated using LDAK5[30] with default parameters. For each trait, $h^2$ was estimated by fitting an additive linear model with a random effect term whose variance was obtained from the GRM, and added age, sex, BMI, 6 genetic PCs and head angles as covariates.

An LD-pruned set of 93,328 autosomal SNPs was used to estimate European, African and Native American ancestry proportions using supervised runs of ADMIXTURE[75]. Reference parental populations included in the ADMIXTURE analyses consisted of Africans (101 Yoruba in Ibadan, Nigeria) and Europeans (107 Iberian Population in Spain) from 1000 Genomes Phase 3 and 125 selected Native Americans[76].

GWAS was conducted on the 301 ILD phenotypes using PLINK v1.9. Sex, age, BMI, three head angles (yaw, pitch, roll) and the first 6 genetic PCs were included as covariates. The Q-Q plots for all traits showed no sign of inflation, and the genomic inflation factor (lambda) of all traits was close to 1 with the maximum value of 1.074 and median value of 1.048, which indicate that appropriately controls for population stratification had been taken care of. Q-Q plots and Manhattan plots[77] of all 301 ILDs are available via figshare https://doi.org/10.6084/m9.figshare.19728916[78].

Multiple testing in the primary GWASs was corrected by estimating the FDR threshold with the Benjamini–Hochberg procedure. The FDR significance threshold was calculated to adjust for the total number of tests (M), which is a product of the total number of SNPs and the total number of phenotypes. Using the classical BH-FDR method[79] to correct for M = 1,342,638,980 tests, the adjusted genome-wide significance threshold was $1.823 \times 10^{-6}$. An alternative FDR approach, used in Xiong et al.[17] and developed in Li et al.[79], is to calculate the effective number of independent tests ($M_{eff}$). For the ILD phenotypes, an eigenvalue decomposition of their correlation matrix was used to calculate the effective number of independent phenotypes, 31.53. For the SNPs, LD pruning was used on the imputed genotypes to calculate the number of effective number of independent SNPs, 1,062,091. Therefore the effective number of independent statistical tests was their product, $M_{eff} = 33,484,596$. With this approach, the adjusted genome-wide significance threshold was very similar, $1.825 \times 10^{-6}$. Both are more lenient than the commonly used GWAS genome-wide significance threshold ($5 \times 10^{-8}$). Therefore, we continued to use the conventional GWAS threshold $5 \times 10^{-8}$ as the genome-wide significance threshold, as this will satisfy the conventional threshold as well as the FDR criteria. However, the genomic regions whose P-value are in between the FDR threshold ($1.823 \times 10^{-6}$) and the commonly used GWAS threshold ($5 \times 10^{-8}$) were presented in Supplementary Table 8.

To group SNP-based GWA results across all analyses based on linkage disequilibrium (LD) between SNPs, we conducted clumping in PLINK v1.9 on the combined output file of all GWA analyses. We used 0.1 for LD threshold, and 1000 Kb for the physical distance threshold, which in total resulted in 62 clumps. To further determine if each clump is independent, we conducted conditioned analyses on the signals physically close to each other. All covariates used in the original GWA analysis were also added in the conditional GWAS. All signals with a conditioned P-value greater than $5 \times 10^{-8}$ were merged with their neighboring signals.

Conditional GWAS was also carried out to test if a signal detected here had been reported previously. We firstly picked out signals that fall on the chromosome bands that have been reported. Amongst 42 regions we detected, 16 fell on an entirely new chromosome band that was not reported to be associated with facial features. We then have conducted the conditional analysis on totally 93 SNPs across 26 regions. We gathered all reported SNPs in each chromosome band and added those reported SNPs into the regression models of corresponding SNPs of the same chromosome band in our results. If P-value obtained was above the suggestive significant threshold ($1 \times 10^{-5}$), this signal would be regarded as a reported signal, and conversely, it would be regarded as a new signal. Details of the results from conditional analyses could be seen in Supplementary Table 9.

In the replication analysis, 76 P-values were available for 27 novel associated regions across 4 separate datasets. After correction of multiple testing with BH-FDR, the combined significance threshold in the replication cohorts was 0.0293.

**Genome-wide association analyses and correction for population stratification**. To verify that population stratification is properly accounted for in our GWAS, we tested several alternative approaches and models (Supplementary Figs. S8–S10):

i. The primary GWAS model described above, implemented using PLINK and including the SNP genotype and genetic PCs (representing whole-genome ancestry);

ii. The same GWAS model using PLINK as in (i) but without genetic PCs, in order to assess the extent of inflation when no adjustment for population structure is included in the model;

iii. A mixed-effect regression model implemented in GCTA[80], which uses a GRM (calculated as above using LDAK) instead of genetic PCs;

iv. An extension of the GCTA mixed effects model, implemented in GENESIS[81,82], using both a relatedness matrix (estimated using KING-robust to model recent kinship), and genetic PCs to model population substructure;

v. A model proposed by Atkinson et al.[83] (TRACTOR) which instead of using the SNP genotypes directly, uses the SNP coding on three different local ancestry backgrounds. We examined three models: one using only genetic PCs (i.e. global ancestry), one using only local ancestry (obtained here by RFMix), and one with both local ancestry and genetic PCs as covariates.

vi. An alternative to TRACTOR called SNP1 (proposed by Hou et al.[64]), which for each SNP genotype uses the local ancestry estimates at that location as covariates. We tested two models: one using both local ancestry and genetic PCs as covariates, and one only using local ancestry but not PCs;

We first contrasted GWAS results for the 148 facial distances showing significant association in the primary PLINK analyses. We ran GCTA and GENESIS on the exact same imputed genome-wide dataset used in the PLINK analyses. For SNP1 and TRACTOR analyses were performed only on the chip data, as local ancestry estimates are only available for genotyped (not imputed) data. To compare results across all analysis approaches and models, the genomic inflation factor (lambda) was calculated for each GWAS using the chip SNP data (Supplementary Fig. 8 and Supplementary Table 10). Examining the distribution of lambda values, we observe that, in the absence of correction for population stratification (PLINK with no PCs, nor GRM, nor local ancestry), there is a marked inflation (Supplementary Fig. 8). However, we find that with any form of whole-genome-based adjustment (with PLINK, GCTA or GENESIS) this inflation is properly controlled, as the lambdas are very close to 1 (as previously observed[64,74,84]). Supplementary Fig. 8 also shows that local ancestry correction (by itself) is not sufficient to correct for stratification in both the SNP1 and TRACTOR analyses (max lambda being 1.7 and 2.4 respectively). Furthermore, genetic PCs on their own are also not sufficient to correct for stratification using TRACTOR (median lambda of 28.5). The best population stratification correction for SNP1 and TRACTOR is obtained when both genomic PCs and local ancestry are incorporated in the models (and we focused on these models in comparisons below).

We next compared -log(P-values) for 42 index SNPs identified in the primary PLINK analyses (across the 148 associated distances), with the values obtained for these SNPs using GCTA and GENESIS. These three approaches produce very similar results, a scatterplot of -log(P-values) showing points that lie close to the diagonal (Supplementary Fig. 9A), matching previous findings[24,64,82,84,85]. We could not perform a similar comparison involving SNP1 or TRACTOR, since certain of the index SNPs identified in the primary PLINK analyses were imputed. We therefore also tested 151 chip-genotyped SNPs (that are significant and in LD, $r^2 > 0.1$, with the 42 PLINK index SNPs) using GCTA, GENESIS, SNP1, and TRACTOR (Supplementary Fig. 9B). We observe that SNP1 and TRACTOR often have a reduced power, relative to the three models incorporating a global ancestry correction (PLINK, GCTA, and GENESIS). This is seen, for instance, for the well-established EDAR[1,2,8] and RUNX2[1,2,4,8,11,15,55,86] gene regions (these SNPs are highlighted in Supplementary Fig. 9B). To further compare power across PLINK, GCTA, GENESIS, SNP1, and TRACTOR, Supplementary Fig. 10 shows violin plots for the six well-established regions that include genotyped associated SNPs in the CANDELA data (taken from Table 1: EDAR[1,2,8], RUNX2[1,2,4,8,11,15,55,86], SLC24A5[2], FOXD1[31,87], GLI3[1,15,88,89], and DCHS2/SFRP2[1,2,4,15,90]). In all six cases, the global ancestry-corrected models (PLINK, GCTA, and GENESIS) have the highest power, followed by SNP1, while TRACTOR has the lowest power. This is particularly noticeable for EDAR and SLC24A5, two instances in which the index SNPs have fully differentiated allele frequencies between populations participating in the admixture (i.e. alternative alleles are fixed in Europeans, Native Americans or Africans). In these two cases, the local Native American ancestry component is nearly identical to the SNP genotype, leading to SNP1 and TRACTOR suffering from high collinearity and resulting in a nearly total loss of power for these two approaches (Supplementary Fig. 10A, C).

Altogether, the analyses above agree with theoretical and simulation studies[64,74,84], which show that, in the absence of close kinship, genetic PCs are sufficient to account for population substructure in GWAS of admixed populations. In such cases, including genetic PCs in the analyses (as implemented in PLINK) produce identical results to using a mixed-effect regression model, which incorporate a genetic relatedness matrix instead of genetic PCs (as implemented in EMMAX[85] or FastLMM[24]). Regarding local ancestry correction approaches, other than collinearity, the drop of power probably stems from effect sizes generally not being sufficiently different between ancestry components to reach genome-wide significance in each ancestry component[65]. Since TRACTOR has three degrees of freedom, with three ancestry-specific SNP components, this approach can be more powerful only when there is substantial heterogeneity in the effect size of SNP across ancestry components[66]. In our case, the trade-off between the scarcity of variants with ancestry-specific effect sizes and that of variants with effect size shared across ancestries appears to be a handicap for TRACTOR.

### Mouse analyses.
We reanalyzed genome-wide SNP and craniofacial data obtained for a published GWAS in outbred mice[34]. Coordinates for 44 landmarks (17 pairs of symmetric landmarks and 10 landmarks on the median plane), along with genotypes at 70k SNPs for 692 mice were kindly provided by Luisa Pallares. We performed a full generalized Procrustes analysis with object symmetry[91], and the phenotypic variation was modeled on the basis of the 67 non-null PC. We applied a multivariate mixed model not used in the original analysis of these data[34]. The

original mouse GWAS was done on each shape PC[34]. However, this approach has maximum power when an allele effect is sufficiently strong to structure the overall shape variation. With geometric morphometrics on skull shape, this is unlikely and a multivariate GWAS is preferable. Such an approach is nevertheless computationally challenging when a linear mixed model (mvLMM) based on the genomic relatedness matrix is used on a very high dimensional trait such as skull shape (here 67 non-null dimensions). We therefore approximated this mvLMM by modeling the covariance matrices of this linear mixed model with two blocks (including skull centroid size as covariate). The first block models the genetic and environmental covariances of the first 10 PCs (62% of the total shape variance) altogether, while only the variances for the next 57 traits were modeled as the second block (i.e. the covariances among these PCs as well as with the other block were set to 0). This approach gains from the modeling of the genetic correlations between the main PCs while maintaining a lower dimensionality cost than in the full multivariate model. Association between a SNP and craniofacial shape was tested based on Pillai trace statistics obtained from the multivariate regression between the corrected allele dosage and corrected PC scores. A FDR was computed based on 100 permutations of corrected PC scores following the approach of Nicod et al.[92] and used to identify SNPs exceeding a FDR threshold of 5%.

### Neanderthal introgression analyses.
These analyses focused on a 1 Mb window around the ATF3 gene in 1q32.3. Imputed genotypes of all samples for 4311 SNPs in this region were first phased using SHAPEIT4 (with default parameters). The haplotypes obtained were re-phased using low-density chip-genotype data previously phased using RFMix (v1)[93]. This two-step-phasing is expected to be more accurate and also aligns phases with the local ancestry estimates obtained by RFMix, hence allowing to determine on which ancestral background the archaic tracts are found. The rephased data was merged with data for "Altai" Neanderthal and the 108 YRI samples from 1000GP3 (used as archaic and modern reference data, respectively) and then filtered. In brief, variants from the 1 Mb window were retained if they: (i) had a read depth ≥20 in "Altai" Neanderthal, (ii) survived the PASS filter in both the "Altai" Neanderthal and 1000GP3 VCFs, (iii) the same ancestral and derived alleles were reported in the two VCFs, and (iv) the ancestral allele is present in our data. This filtering resulted in 3231 SNPs being retained. The introgression scan, on the filtered data, was carried out using the hidden-Markov model implemented in admixtureHMM[41], considering only tracts called with >99% confidence and that were >10 Kb. This identified 798 introgression tracts, with an average length of 127 Kb.

We performed association testing through (archaic) admixture mapping, that is, we first recoded genotypes based on the number of archaic alleles (0,1 or 2) and merged consecutive SNPs with a similar distribution of genotypes across individuals, allowing a maximum of 1% genotype change across individuals from one SNP to the next. Filtering for a minimum archaic tract frequency of 1%, led to a total of 103 introgressed segments being retained. We then tested for phenotypic association using the same linear model as for the GWAS. The Benjamini–Hochberg's FDR significance threshold equals $4.9 \times 10^{-4}$.

### Neanderthal and modern human skull comparison.
Distance D117 (between landmarks 12/nasion and 18/subnasal) measured in the CANDELA individuals corresponds to the cranial distance measured between nasion and subspinale landmarks (i.e. nasal height, in Howells' system of cranial measurement[94]). We extracted nasal height from the measurements obtained by Weaver and Stringer[95] on 10 Neanderthal specimens (Amud 1, Forbes' Quarry, Guattari 1, La Chapelle-Aux-Saints, La Ferrassie 1, Saccopastore 1 and 2, Saint-Césaire, Shanidar 1, Shanidar 5). For comparison with modern humans, we extracted nasal height measured in skulls from 484 Africans, 317 Europeans and 389 Native Americans (males and females were balanced for each region), from Howells' online database (http://web.utk.edu/~auerbach/HOWL.htm)[94]. To illustrate the nasal height difference between modern human and Neanderthal skulls, we compared a Native American and the Amud 1 Neanderthal (41Kya). The 3D image of the Native American skull was obtained from the collection of the División de Antropología, Museo de La Plata, Argentina (skull from Chubut Province, DA-MLP-1082). The Amud 1 3D image was obtained from the MorphoSource repository (www.morphosource.org): Darwin Core triplet: du:ea:CCC08 Homo neanderthalensis; ID Media 000005749: Cranium [Mesh] [Etc]. MorphoSource Archival Resource Key (ARK) identifier: ark:/87602/m4/M5749.

### Annotation of SNPs in FUMA (functional mapping and annotation).
A subset of GWAS summary statistics including only significant SNPs ($P < 5 \times 10^{-8}$) and a pre-defined lead SNP list obtained after clumping in Plink v1.9 were loaded to FUMA[44]. SNP2GENE was processed to identify independent SNPs ($r^2 < 0.6$) and candidate SNPs. Candidate SNPs are the SNPs in LD of one of the independent significant SNPs, which includes non-GWAS tagged SNPs extracted from 1000 genomes reference panel. Implemented tool ANNOVAR was used to annotate the functional consequences on gene function on the independent SNPs and candidate SNPs. The website indicates that ANNOVAR uses all annotated transcripts in Gencode collection lifted up to hg19, and has its own prioritization criteria to report the most deleterious function. Only prioritized annotations are used for those SNPs.

**Gene Ontology (GO) analysis and transcription patterns in newly associated regions**. We used Metascape (http://metascape.org/) to carry out a GO analysis[96] of genes nearest to the index SNPs of the novel associated regions (if an index SNP was in two genes, both genes were retained in the analysis) (Supplementary Table 4). To examine patterns of transcription in the vicinity of index SNPs for the novel regions identified here, we contrasted the CNCC RNA-seq data from the study of Prescott et al.[45] to that obtained by the ENCODE[46] project for 318 different cell types (Supplementary Table 11). Of the 33 newly associated regions, overlapping transcripts in the CNCC RNAseq data have been reported for 26, and only these could therefore be tested. For consistency with the CNCC data, we applied variance-stabilizing transformation (VST) to the ENCODE data (using DESeq2). The higher transcription levels in CNCCs, relative to the ENCODE data, was tested using a Student's $t$ test, with a Benjamini–Hochberg's FDR threshold ($p < 0.034$).

**Reporting summary**. Further information on research design is available in the Nature Portfolio Reporting Summary linked to this article.

## Data availability

Raw genotype or phenotype data cannot be made available due to restrictions imposed by the ethics approval. Summary statistics obtained during the current study have been deposited at GWAS central and is available at the URL http://www.gwascentral.org/study/HGVST5029. All produced Manhattan plot and Q-Q plot are available via figshare https://doi.org/10.6084/m9.figshare.19728916[78]. Supplementary Tables can be found in the Supplementary Data file. All other data are available from the corresponding author on reasonable request. Public data resources used: The Altai Neanderthal genome was downloaded from the website of the Max Planck Institute for Evolutionary Anthropology at http://cdna.eva.mpg.de/neandertal/altai/AltaiNeandertal/VCF/. European cohort summary statistics: https://doi.org/10.6084/m9.figshare.10298396[68]. For the R package FastMan used to draw the Manhattan plot in Fig. 1 and Q-Q plots in Supplementary Materials, see https://github.com/kaustubhad/fastman.

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

## Acknowledgements

Professor Gabriel Bedoya led the CANDELA team in Colombia but passed away during preparation of this manuscript. We thank the volunteers for their enthusiastic support for this research. We also thank Alvaro Alvarado, Mónica Ballesteros Romero, Ricardo Cebrecos, Miguel Ángel Contreras Sieck, Francisco de Ávila Becerril, Joyce De la Piedra, María Teresa Del Solar, William Flores, Martha Granados Riveros, Rosilene Paim, Ricardo Gunski, Sergeant João Felisberto Menezes Cavalheiro, Major Eugênio Correa de Souza Junior, Wendy Hart, Ilich Jafet Moreno, Paola León-Mimila, Francisco Quispea-laya, Diana Rogel Diaz, Ruth Rojas, and Vanessa Sarabia, for assistance with volunteer recruitment, sample processing and data entry. We are very grateful to the institutions that allowed the use of their facilities for the assessment of volunteers, including: Escuela Nacional de Antropología e Historia and Universidad Nacional Autónoma de México (México); Universidade Federal do Rio Grande do Sul (Brazil); 13° Companhia de Comunicações Mecanizada do Exército Brasileiro (Brazil); Pontificia Universidad Católica del Perú, Universidad de Lima and Universidad Nacional Mayor de San Marcos (Perú). We acknowledge M Arfan Ikram, Tamar EC Nijsten, Markus A de Jong, Stefan Boehringer, Myoung Keun Lee, Eleanor Feingold, Mary L Marazita, Lavinia Paternoster, Holly Thompson, Gemma C Sharp, Sarah Lewis, Stephen Richmond, Alexei Zhurov and Luisa Pallares for facilitating access to published datasets. We thank Luisa Pallares and Abraham Palmer for kindly sharing the GWA mouse data. Centre de Calcul Intensif d'Aix-Marseille is acknowledged for granting access to its high performance computing resources. We thank Joanne Cole for providing us the GWAS summary statistics from

one of the African population GWAS studies, which we used as one of our replication panel. We thank the MorphoSource repository (www.morphosource.org) and the División de Antropología, Museo de La Plata (Argentina) for access to the skull 3D images shown in Fig. 3.

Work leading to this publication was funded by grants from: the National Natural Science Foundation of China (#31771393), the Scientific and Technology Committee of Shanghai Municipality (18490750300), Ministry of Science and Technology of China (2020YFE0201600), Shanghai Municipal Science and Technology Major Project (2017SHZDZX01) and the 111 Project (B13016), the Leverhulme Trust (F/07 134/DF), BBSRC (BB/I021213/1), the Excellence Initiative of Aix-Marseille University - A*MIDEX (a French "Investissements d'Avenir" programme), Universidad de Antioquia (CODI sostenibilidad de grupos 2013- 2014 and MASO 2013-2014), Consejo Nacional de Desenvolvimento Científico e Tecnológico, Fundação de Amparo à Pesquisa do Estado do Rio Grande do Sul (Apoio a Núcleos de Excelência Program), Fundação de Aperfeiçoamento de Pessoal de Nível Superior and the National Institute of Dental and Craniofacial Research (R01-DE027023; U01-DE020078; R01-DE016148; X01-HG007821), Santander Research & Scholarship Award. B.B. is supported by a doctoral scholarship from Ecole Doctorale 251 Aix-Marseille Université.

## Author contributions

B.B., M.F.G., J.M.R., J.C.C.D., M.H., V.V., V.G., C.J., W.A., R.B., P.E.M., M.S.Q., J.G.V., H.V.R., C.C.S.C., T.H., V.R., R.G.J., L.S.F., M.C.B., V.A.A., S.C.Q., C.G., G.P., W.R., and F.R. contributed to volunteer recruitment or data collection. Q.L., J.C., K.A., N.N., P.F., M.E.D., A.G., S.S.P., M.R.K., S.Wu., and S.D. performed analyses. K.A., N.N., and R.G.J. provided guidance on aspects of study design. A.R.L., N.N., and S. Wang obtained funding or provided access to resources. A.R.L. and K.A. designed the project. A.R.L., Q.L., K.A., and J.C. wrote the paper with input from co-authors. A.R.L. coordinated the study.

## Competing interests

The authors declare no competing interests.

## Additional information

[1]Ministry of Education Key Laboratory of Contemporary Anthropology and Collaborative Innovation Center of Genetics and Development, School of Life Sciences and Human Phenome Institute, Fudan University, Yangpu District, Shanghai 200438, China. [2]CAS Key Laboratory of Computational Biology, Shanghai Institute of Nutrition and Health, University of Chinese Academy of Sciences, Chinese Academy of Sciences, 320 Yue Yang Road, Shanghai 200031, China. [3]Aix-Marseille Université, CNRS, EFS, ADES, Marseille 13005, France. [4]División Antropología, Facultad de Ciencias Naturales y Museo, Universidad Nacional de La Plata, La Plata, República Argentina. [5]Consejo Nacional de Investigaciones Científicas y Técnicas, CONICET, Buenos Aires, República Argentina. [6]Departamento de Tecnología Médica, Facultad de Ciencias de la Salud, Universidad de Tarapacá, Arica 1000000, Chile. [7]Laboratorios de Investigación y Desarrollo, Facultad de Ciencias y Filosofía, Universidad Peruana Cayetano Heredia, Lima 31, Perú. [8]Unit of Human Evolutionary Genetics, Institut Pasteur, Paris 75015, France. [9]Division of Vertebrates and Anthropology, Department of Earth Sciences, Natural History Museum, London SW7 5BD, UK. [10]GENMOL (Genética Molecular), Universidad de Antioquia, Medellín 5001000, Colombia. [11]Molecular Genetics Laboratory, National School of Anthropology and History, Mexico City, 14050, Mexico 6600, Mexico. [12]Department of Archaeogenetics, Max Planck Institute for the Science of Human History (MPI-SHH), Jena 07745, Germany. [13]Forensic Science, Faculty of Medicine, UNAM (Universidad Nacional Autónoma de México), Mexico City 06320, Mexico. [14]Unidad de Genomica de Poblaciones Aplicada a la Salud, Facultad de Química, UNAM-Instituto Nacional de Medicina Genómica, Mexico City 4510, Mexico. [15]Scientific Police of São Paulo State, Ourinhos, SP 19900-109, Brazil. [16]Departamento de Genética e Biologia Evolutiva, Instituto de Biociências, Universidade de São Paulo, São Paulo, SP 05508-090, Brazil. [17]Departamento de Genética, Universidade Federal do Rio Grande do Sul, Porto Alegre 90040-060, Brazil. [18]Instituto Patagónico de Ciencias Sociales y Humanas, Centro Nacional Patagónico, CONICET, Puerto Madryn U9129ACD, Argentina. [19]School of Mathematics and Statistics, Faculty of Science, Technology, Engineering and Mathematics, The Open University, Milton Keynes MK7 6AA, United Kingdom. [20]Instituto de Alta Investigación, Universidad de Tarapacá, Arica, Arica 1000000, Chile. [21]Biogéosciences, UMR 6282 CNRS, Université de Bourgogne, Dijon 21000, France. [22]EPHE, PSL University, Paris 75014, France. [23]Department of Genetics, Evolution and Environment, and UCL Genetics Institute, University College London, London WC1E 6BT, UK. [24]These authors contributed equally: Qing Li, Jieyi Chen. [25]These authors jointly supervised this work: Kaustubh Adhikari, Andrés Ruiz-Linares.
✉email: kaustubh.adhikari@open.ac.uk; andresruiz@fudan.edu.cn

