## [Peer Review File · Communications Biology]

Reviewers' comments:

Reviewer #1 (Remarks to the Author):

In this manuscript, Li and colleagues show results from automatic landmarking of 2D photographs using the cloud service platform Face++ in >6,000 Latin Americans for 301 inter-landmark distances and detected nominally significant facial features association for 42 genome regions. Furthermore, the authors show that nine regions have been previously reported in several GWAS studies and replicated 26 of the 33 novel regions, including 1q32.3, 3q21.1, 8p11.21, 10p11.1, and 22q12.1, regions which contain candidate genes involved in craniofacial development, and comparison with findings from previous studies confirmed the significance of these regions. Results from this study indicates that 2D photographs might be useful and informative, and automated landmarking of 2D photographs might be an alternative reliable approach for the genetic analysis of facial variation.

Although the authors used data from two additional studies to evaluate their findings, they didn't compare their findings to one of the largest African population studies which is part of the FaceBase Consortium. Available datasets can be found here:

https://www.facebase.org/chaise/recordset/#1/isa:dataset/*::facets::N4IghgdgJiBcDaoDOB7ArgJwMYFM6JAEsIAjdafIpMEAGhCjABCwkcmb9FDAC0kKQBbDoxZtOhKBwBmAaxwBPEAF0AvrVDomZNBQRUa9Ua3Zde-

[IWb4QBwuYpXqiMzfSwALFIVxJKAOQBhACEASQAVAEADQB5P1gATgA2AAYkxzUgA@sort\(release_date::desc::,RID\),](https://www.facebase.org/chaise/recordset/#1/isa:dataset/*::facets::N4IghgdgJiBcDaoDOB7ArgJwMYFM6JAEsIAjdafIpMEAGhCjABCwkcmb9FDAC0kKQBbDoxZtOhKBwBmAaxwBPEAF0AvrVDomZNBQRUa9Ua3Zde-IWb4QBwuYpXqiMzfSwALFIVxJKAOQBhACEASQAVAEADQB5P1gATgA2AAYkxzUgA@sort(release_date::desc::,RID),) and the authors should check the following studies to evaluate the novelty of their findings: <https://doi.org/10.1016/j.xhgg.2021.100082>,

<https://doi.org/10.1371/journal.pgen.1006174>, <https://doi.org/10.1534/genetics.116.193185>,

<https://doi.org/10.1371/journal.pgen.1009695>

Limitations of 2D technology, limitations compared to 3D studies should be included

It is not clear which human genome reference build the authors used for this study

Lines 101-102: please include the exact number of novel associations

Lines 108-110: please include the number of samples included in this study

Line 120: not clear what "ICC" means, it's not described in previous sections/paragraphs.

Line 164: "After quality control filters", it is not clear whether the authors refer to the previous section or additional quality control steps, please elaborate what is meant by "quality control filters".

Lines 451-452: not clear how the authors selected these 34 landmarks, please provide more details

Line 456: please provide the number of duplicates removed from this study

Line 475: please explain briefly why 1.82×10^{-6} wasn't a good threshold for downstream analyses

Lines 502-503: this sentence is not clear

Lines 515-571: the filtering step is not very detailed, please include more information

Reviewer #2 (Remarks to the Author):

Visual traits have been the source of fascination since the dawn of time and family resemblances is perhaps the first acknowledged heritable trait. Qing Li et al. have submitted the result of genetic association analyses that would explain part of the heritability of facial features. This study combines numerically strong (in relative terms) sample sizes of multiethnic origin and an automated phenotyping approach that appears superior to analogous methodologies previously that are already in the published literature. Landmarks were digitally determined from 2D frontal photographs, and 301 unique inter-landmark distances were obtained and analyzed through linear regression models that used SNP as predictors, after adjusting from common and expected confounders. The authors used a heavily admixed population at the discovery stage and then use two monoethnic samples (Chinese

and European) to replicate these findings. This is a good research objective, and the general readership would be interested in such findings.

While the manuscript is of interest, it often lacks clarity. There are clarity-related issues in the way the manuscript is laid out and the methodology is often difficult to follow. The instances listed below exemplify what, in the opinion of this reviewer, are the main issues with the manuscript.

The authors mention that they used linear regression methods in a multiethnic and cohort with high levels of admixture. To some readers, this approach would be the mother of all population stratification problems. Adjusting for a limited number of genetic principal components, or indeed any number of PCs is not a way to tackle these issues. The authors are also not reporting any genomic inflation values and the reader, like the reviewer, may be wondering whether this issue is properly taken care of. In practice this would mean that a trait that is particularly prominent in Native Americans, would be disproportionately associated with all, if not most SNP markers that are more common in that particular ethnic group.

Another instance of vague methodology is the authors description of multiple testing correction. In Line 471 they mention BH-FDR. It is unclear whether that methodology was applied genome-wide or to the stage where SNPs were replicated (if so, was this applied to SNPs selected after conditional analyses?). In the next line the authors seem to settle for the conventional GWAS level of association. It also appears as if the authors used FDR to adjust for multiple testing not only across different loci but also different traits, which is a debatable way to deal with the phenotypic complexity of the human face. Although the distances are doubtlessly correlated, there are several better ways to calculate the degree of phenotypic independence and estimate a multiple testing correction factor.

The methodology used for the gene expression analysis is also unclear and not convincing. It seems that they compared the expression of a number of transcripts that were already known to be expressed in the neural crest cells ("among all 118 genes annotated to 42 significant genomic regions, we found that 103 genes could express in CNCCs.") with the rest (N=55,779). If so, there is perhaps little surprise that the results were significant, since we know that in any given tissue or cell, only 10% of the transcripts are expressed beyond trace level. The authors are reporting that a group of genes, collectively, tend to be expressed more than some other group – even if true, it is hard to see why this information would be of any practical interest. The individual expression levels of single genes are not reported. Once more, there are other approaches to study tissue enrichment of GWAS results and the authors may want to consider using more established tools.

The authors rightly describe replication of previously published regions of association. While there is some genuine interest in knowing how well these results align with previous works, the very detailed presentation of essentially known associations and genes over three pages of text appears a little excessive. In fact, known regions are described more at length than the novel results.

The authors report correlation of distances with sex. Although not explicitly mentioned, the impression was that the Pearson's r coefficients were reported. A Pearson correlation between a binary and a quantitative variable are rather difficult to interpret, and one is never sure what it means. Worth considering a point biserial correlation?

The authors may want to improve on the methodological descriptions; currently imprecise language is used at some frequency, for example "we firstly computed a genomic relationship matrix after gathering all individuals who appeared in at least one trait", "expression analysis for significant SNPs" (SNPs don't express), then they mention testing expression against a "group that includes the other 55,779 genes", which is a number that appears too high for the human genome (maybe use "gene expression" and "transcripts" instead?).

Reviewer #3 (Remarks to the Author):

This study presents a GWAS for facial variation in the CANDELA cohort, where the authors found a total of 42 genomic regions associated with facial features from 2D photographs. Overall, the

manuscript is well written and I have the following comments to further clarify the study:

- 1) There is a lot of data in the study and it can be difficult to follow the analyses at times. It is unclear how many GWASes were conducted. Additionally, summaries of these analyses, such as Manhattan plots and QQ plots should be included.
- 2) It is unclear how the results from this analysis differ from the results from the analysis that used manual landmarks (or other studies of facial variation). For the results that were nominal significant in this analysis, what were their p-values in the previous analysis? This should be added to the study.
- 3) Pg 5: It is unclear how European/Native American ancestry is defined here. More details are needed
- 4) Pg 6: It is unclear how both a p-value threshold and an FDR threshold were applied. Additionally, the exact threshold for the FDR cut-point is undefined. More details are needed
- 5) Pg 9: It is again unclear what the P-value and FDR threshold is and how these were applied.
- 6) Pg 9: It is unclear how the candidate genes were selected for this analysis. More details are needed
- 7) Pg 9: It is unclear how the archaic introgression analysis was conducted, how these regions were defined, and how they were chunked into 140 segments. More details are needed.
- 8) Pg10: It is unclear what multivariate mixed model was used. More details are needed
- 9) Pg 10: It is unclear what the strong association was between. More details are needed
- 10) Pg 11: Besides a brief mention that the manual landmarking and the landmarks provided by Face++ were correlated, there was no evidence presented that this software is reliable and this statement should be removed from the conclusion.
- 11) Pg13: It is unclear if there was any additional quality control for imputation probability or MAF was conducted after imputation. This should be added to the methods.
- 12) Pg 13: The statement on multiple test correction is unclear was a FDR threshold used or was a Bonferroni correction? What analysis was this threshold used for?
- 13) Pg 14: the statement on FDR and combined significance is unclear and it is not apparent what analysis this is for.

RESPONSE THE REVIEWERS' COMMENTS.

Our replies below are shown in **bold**. Related changes have been highlighted in the manuscript file (quotes are shown here in *italic*).

In addition to the changes made to the manuscript in response to the reviewers' comments we have:

1. Combined Supplementary Figure S2 with Supplementary Table S1.
2. Reorganized parts of the results subsections. Relabeled the number of tables, figures and supplementary tables and figures.
3. Moved Figure 2 into supplementary figures.
4. Added two authors to author list of the revised manuscript, and therefore the institution is re-numbered. These two authors contributed in the comparison between Face++ and Dlib included in the revision.
5. Updated Figure 1B. Two colours instead of three are now used to indicate replication and novel findings in this study, and we have revised the figure legend accordingly.
6. Updated Figure 2D in this revised version. The y-axis label has now changed from "logP" to "-log(P-value)".
7. Updated Figure 4. The index SNP is now indicated in the revised version of the figure. Figure legend was also revised.

Reviewer #1 (Remarks to the Author):

In this manuscript, Li and colleagues show results from automatic landmarking of 2D photographs using the cloud service platform Face++ in >6,000 Latin Americans for 301 inter-landmark distances and detected nominally significant facial features association for 42 genome regions. Furthermore, the authors show that nine regions have been previously reported in several GWAS studies and replicated 26 of the 33 novel regions, including 1q32.3, 3q21.1, 8p11.21, 10p11.1, and 22q12.1, regions which contain candidate genes involved in craniofacial development, and comparison with findings from previous studies confirmed the significance of these regions. Results from this study indicates that 2D photographs might be useful and informative, and automated landmarking of 2D photographs might be an alternative reliable approach for the genetic analysis of facial variation.

1. Although the authors used data from two additional studies to evaluate their findings, they didn't compare their findings to one of the largest African population studies which is part of the FaceBase Consortium. Available datasets can be found here:

the following studies to evaluate the novelty of their findings:

<https://doi.org/10.1016/j.xhgg.2021.100082>,
<https://doi.org/10.1371/journal.pgen.1006174>,
<https://doi.org/10.1534/genetics.116.193185>,
<https://doi.org/10.1371/journal.pgen.1009695>

We appreciate the reviewer's overall assessment of our work. We agree with the reviewer that it would be useful to compare our results with African population studies (although the African component of our sample is relatively small). We have now added two columns (Cols L-M) to Supplementary Table S4 (including the SNP and CNV GWAS P-values reported in an African population[1, 2]). As we added these two African datasets for assessment of replication, we recalculated the FDR threshold of replication P-value (we therefore added details on these African data in the Methods section in lines 379-383). We also checked the other four studies mentioned by the reviewer, but none of our novel signals were detected in them.

2. Limitations of 2D technology, limitations compared to 3D studies should be included

We have now added a discussion of the limitations of 2D imaging technology compared to 3D imaging technology in the Conclusion section in lines 327-332.

“Naturally, landmarks (and measurements extracted from 2D photographs) have obvious limitations relative to traits extracted from 3D facial scans. The 2D landmarks are essentially projections of their 3D locations onto a plane, and therefore measurements obtained from 2D landmarks are distorted versions of 3D-based measurements. Therefore, in the case of facial photographs, 2D measurements are influenced by head position when taking the photographs; for this reason, Face++ estimates three head positioning angles (yaw, pitch, roll).”

3. It is not clear which human genome reference build the authors used for this study

We thank the reviewer for pointing out this omission. We have now specified the build used in the Methods section in line 392:

“Human genome reference assembly GrCh37/hg19 was used.”

4. Lines 101-102: please include the exact number of novel associations
**We now provide the exact number of novel associations in line 101:
*“In addition, we identify 33 novel associations with facial traits.”***
5. Lines 108-110: please include the number of samples included in this study
We now mention the exact number of samples included in the study in lines 108-110:

“The 6,486 individuals examined here are part of the CANDELA cohort, collected in five Latin American countries¹⁷. This cohort has been previously examined in several GWASs of physical appearance^{1, 15, 16, 18, 19}”

6. Line 120: not clear what “ICC” means, it’s not described in previous sections/paragraphs.
**We now spell out “ICC” the first time it is used in line 121:
“Interclass correlation coefficients (ICCs) between landmarks placed manually (taken as the ground truth), by Face++, or Dlib, were mostly >0.90”**
7. Line 164: “After quality control filters”, it is not clear whether the authors refer to the previous section or additional quality control steps, please elaborate what is meant by “quality control filters”.
We now provide full details on the quality control filters in the Methods section in lines 388-392 and lines 410-413.
8. Lines 451-452: not clear how the authors selected these 34 landmarks, please provide more details
**We now expand the text explaining how these landmarks were selected in lines 413-421 and also added a column to Supplementary Table S2, which includes citations that use these landmarks.
“We focused on 34 landmarks corresponding mostly to well-defined anatomical landmarks of common usage^{23, 24} including previous GWAS studies (Figure 1A, Supplementary Table S2)^{4, 5, 16, 22, 25}. Specifically, 28 out of 34 landmarks are well-defined anatomical landmarks, while the other six landmarks (these are more commonly referred to as “semi-landmarks” or “pseudo-landmarks” in the physical anthropology literature¹, but to simplify the presentation, are referred to collectively as “landmarks” too) are on important locations such as the end of a contour, which would allow us to capture facial features that we are interested in, including face width and eyebrow size (Supplementary Table S2).”**
9. Line 456: please provide the number of duplicates removed from this study
**We now have provided the exact number of duplicates removed from the study in line 424:
“After removing 260 duplicates, 301 distances (labeled as ‘D’ followed by a number, Supplementary Table S3) were retained for the GWAS.”**
10. Line 475: please explain briefly why 1.82×10^{-6} wasn’t a good threshold for downstream analyses
**We did not use 1.82×10^{-6} because this threshold is more lenient than the commonly used value of 5×10^{-8} . We chose a stricter threshold to reduce false positives. This information is in lines 492-498.
“With this approach, the adjusted genome-wide significance threshold was very similar, 1.825×10^{-6} . Both are more lenient than the commonly used GWAS genome-wide significance threshold (5×10^{-8}). Therefore, we continued to use the conventional GWAS threshold 5×10^{-8} as the**

genome-wide significance threshold, as this will satisfy the conventional threshold as well as the FDR criteria. However, the genomic regions whose P-value are in between the FDR threshold (1.823×10^{-6}) and the commonly used GWAS threshold (5×10^{-8}) were presented in Supplementary Table S5.

11. Lines 502-503: this sentence is not clear
We have now removed this sentence.

12. Lines 515-571: the filtering step is not very detailed, please include more information

We have specified the 4 filters used for selecting variants in the introgression scan. In brief, these filters are: read depth, SNP quality, consistency of ancestral/derived allele information and consistency of polymorphism. This information is in lines 546-551.

“In brief, variants from the 1Mb window were further retained if they complied with all these filters: (i) read depth ≥ 20 in the “Altai” Neanderthal sample, (ii) filter PASS in both the “Altai” Neanderthal and 1000GP3 VCFs, (iii) same ancestral and derived alleles reported in these 2 VCFs, and (iv) the reported ancestral allele is among the 2 alleles in our data.”

Reviewer #2 (Remarks to the Author):

Visual traits have been the source of fascination since the dawn of time and family resemblances is perhaps the first acknowledged heritable trait. Qing Li et al. have submitted the result of genetic association analyses that would explain part of the heritability of facial features. This study combines numerically strong (in relative terms) sample sizes of multiethnic origin and an automated phenotyping approach that appears superior to analogous methodologies previously that are already in the published literature. Landmarks were digitally determined from 2D frontal photographs, and 301 unique inter-landmark distances were obtained and analyzed through linear regression models that used SNP as predictors, after adjusting from common and expected confounders. The authors used a heavily admixed population at the discovery stage and then use two monoethnic samples (Chinese and European) to replicate these findings. This is a good research objective, and the general readership would be interested in such findings.

While the manuscript is of interest, it often lacks clarity. There are clarity-related issues in the way the manuscript is laid out and the methodology is often difficult to follow. The instances listed below exemplify what, in the opinion of this reviewer, are the main issues with the manuscript.

1. The authors mention that they used linear regression methods in a multiethnic and cohort with high levels of admixture. To some readers, this approach would be the mother of all population stratification problems. Adjusting for a limited number of genetic principal components, or indeed any number of PCs is not a way to tackle these issues.

The authors are also not reporting any genomic inflation values and the reader, like the reviewer, may be wondering whether this issue is properly taken care of. In practice this would mean that a trait that is particularly prominent in Native Americans, would be disproportionately associated with all, if not most SNP markers that are more common in that particular ethnic group.

We acknowledge this criticism in that the population stratification problem had not been properly discussed. Working on the comments of Reviewer 1, we added additional details in the Methods section, where we clarify that related individuals were excluded prior to the GWAS analysis, and therefore in absence of recent kinship the genetic PCs should be sufficient to control for population structure. Nonetheless, to demonstrate this, we now include additional GWAS analyses using the program GENESIS[3], which uses two components to model population structure. It is a further extension of the mixed effects model, using two components: a relatedness matrix estimated using KING-robust to model recent kinship, and genetic PCs to model population substructure. We compare the results from GENESIS to the classical GWAS analysis to show that population structure was already properly addressed. This information is in lines 465-480.

“It was seen in theoretical and simulation studies⁷⁸ as well as in a previous CANDELA GWAS using a mixed-effect regression model implemented in FastLMM¹⁸, that in the absence of close kinship, genetic PCs are sufficient to account for population substructure in GWASs. Nonetheless, to further verify that the population stratification problem is properly addressed in our study, we replicated our analyses with GENESIS⁷⁹. It is a further extension of the mixed effects model, using both components: a relatedness matrix estimated using KING-robust to model recent kinship, and genetic PCs to model population substructure. We ran GENESIS on the exact same datasets used in the ordinary PLINK GWAS (i.e. excluding closely related samples), and compared the two sets of results, which were nearly identical. Some comparisons between the results obtained by two software was shown in Supplementary Figure S8. First, we picked a representative trait which has the most significant signal (D198) to show the whole-genome imputed GWAS results with GENESIS against the PLINK results as a scatterplot, with all the points lying along the diagonal. Secondly, for the 42 index SNPs we detected, and for all the traits (including non-significant ones), we perform the same comparison. The genomic inflation factors from the two analyses were also nearly identical.”

In addition, we now report the maximum inflation factor and median inflation factor across all traits in the Method section in lines 460-464 and include all Manhattan plots and QQ plots for all traits (which was also recommended by Reviewer #3):

“The Q-Q plots for all traits showed no sign of inflation, and the genomic inflation factor (λ) of all traits was close to 1 with the maximum value of 1.074 and median value of 1.048, which indicate that appropriately controls for population stratification had been taken care

of. Q-Q plots and Manhattan plots⁷⁷ of all 301 ILDs are available via figshare <https://doi.org/10.6084/m9.figshare.19728916>”

2. Another instance of vague methodology is the authors description of multiple testing correction. In Line 471 they mention BH-FDR. It is unclear whether that methodology was applied genome-wide or to the stage where SNPs were replicated (if so, was this applied to SNPs selected after conditional analyses?). In the next line the authors seem to settle for the conventional GWAS level of association. It also appears as if the authors used FDR to adjust for multiple testing not only across different loci but also different traits, which is a debatable way to deal with the phenotypic complexity of the human face. Although the distances are doubtlessly correlated, there are several better ways to calculate the degree of phenotypic independence and estimate a multiple testing correction factors.

We apologize that our initial text was confusing because adjustment for multiple testing via FDR was being conducted in two completely different situations, but this was not properly described, nor the two FDR procedures in the two situations distinguished. We now make it clear that in the first situation we are applying a multiple testing correction via FDR on the entire set of GWAS p-values. In addition, taking on board the reviewer’s point that such a way of applying FDR is a debatable way to handle the complexity, we also applied an alternative FDR approach used in a previous GWAS study[4], to calculate the effective number of independent phenotypes. This approach and the classic BH-FDR we used originally gave a very similar FDR threshold, and both are larger than the commonly used GWAS threshold. This information is in lines 481-498.

“Multiple testing in the primary GWASs was corrected by estimating the false discovery rate (FDR) threshold with the Benjamini-Hochberg procedure. The FDR significance threshold was calculated to adjust for the total number of tests (M), which is a product of the total number of SNPs and the total number of phenotypes. Using the classical BH-FDR method to correct for M = 1,342,638,980 tests, the adjusted genome-wide significance threshold was 1.823×10^{-6} . An alternative FDR approach, used in Xiong et al., 2019⁴ and developed in Li et al., 2005⁸⁰, is to calculate the effective number of independent tests (Meff). For the ILD phenotypes, an eigenvalue decomposition of their correlation matrix was used to calculate the effective number of independent phenotypes, 31.53. For the SNPs, LD pruning was used on the imputed genotypes to calculate the number of effective number of independent SNPs, 1,062,091. Therefore the effective number of independent statistical tests was their product, Meff = 33,484,596. With this approach, the adjusted genome-wide significance threshold was very similar, 1.825×10^{-6} . Both are more lenient than the commonly used GWAS genome-wide significance threshold (5×10^{-8}). Therefore, we continued to use the conventional GWAS threshold 5×10^{-8} as the genome-wide significance threshold, as this will satisfy the conventional threshold as well as the FDR criteria. However, the genomic regions whose P-value are in

between the FDR threshold (1.823×10^{-6}) and the commonly used GWAS threshold (5×10^{-8}) were presented in Supplementary Table S5.

3. The methodology used for the gene expression analysis is also unclear and not convincing. It seems that they compared the expression of a number of transcripts that were already known to be expressed in the neural crest cells ("among all 118 genes annotated to 42 significant genomic regions, we found that 103 genes could express in CNCCs.") with the rest (N=55,779). If so, there is perhaps little surprise that the results were significant, since we know that in any given tissue or cell, only 10% of the transcripts are expressed beyond trace level. The authors are reporting that a group of genes, collectively, tend to be expressed more than some other group – even if true, it is hard to see why this information would be of any practical interest. The individual expression levels of single genes are not reported. Once more, there are other approaches to study tissue enrichment of GWAS results and the authors may want to consider using more established tools.

We appreciate the reviewer's suggestions, and have made changes to the manuscript to incorporate them. Firstly, the cranial neural crest cells (CNCCs) are basic and important for cranial-facial development. Prescott et al. derived human and chimpanzee CNCCs, and published histone modifications, transcription factors, chromatin accessibility and gene expression as the series GSE70751 in the GEO DataSets. These datasets have been widely used in previous genetic studies on facial morphology. Secondly, as the reviewer mentioned "in any given tissue or cell, only 10% of the transcripts are expressed beyond trace level", the gene expression data used here, including the so-called background genes, could actually be detected in CNCCs. Next, according to the revised candidate gene list of the significant genomic regions, we found that 11 genes are in reported genomic regions and 32 genes are in novel ones. Both groups of genes in reported and novel genomic regions express significantly higher than the group of background genes in CNCCs. Such analysis was similar to the expression analysis used by Xiong et al[4]. The expression levels of the reported and novel group were also similarly high, which means the findings in novel genomic regions made sense. Finally, as the reviewer recommends, we performed the enrichment analysis for all 43 candidate genes annotated to the significant genomic regions by Metascape. We found that the top-level Gene Ontology biological processes were enriched in important growth and development processes, such as GO:0032502: developmental process, GO:0022414: reproductive process, GO:0051179: localization, GO:0023052: signalling and etc. This new result is now included in Supplementary Figure S7 and the descriptions are in lines 302-304.

4. The authors rightly describe replication of previously published regions of association. While there is some genuine interest in knowing how well these results align with previous works, the very detailed presentation of essentially known associations and genes over three pages of text appears a little excessive. In fact, known regions are described more at length than the novel results.

Following the reviewer's suggestion, we have moved the discussion of the replicated regions into Supplementary materials (Supplementary Note 1).

5. The authors report correlation of distances with sex. Although not explicitly mentioned, the impression was that the Pearson's r coefficients were reported. A Pearson correlation between a binary and a quantitative variable are rather difficult to interpret, and one is never sure what it means. Worth considering a point biserial correlation?

We agree with the reviewer that the method of calculating correlation of distances with sex was not specified in the manuscript. We did use Pearson's method, and now have switched it to Spearman's correlation analysis. We totally agree with the reviewer that Pearson's correlation between a binary and a quantitative variable is difficult to interpret. We now have updated our correlation results table (Supplementary Table S3), and revised the text and figures accordingly. We now have also specified the method used for correlation analysis in lines 441-443. *"Phenotype characteristics were explored for gender, aging, BMI, and head angles effect. They were estimated using Spearman's correlation method as some of the variables are not continuous."*

6. The authors may want to improve on the methodological descriptions; currently imprecise language is used at some frequency, for example "we firstly computed a genomic relationship matrix after gathering all individuals who appeared in at least one trait", "expression analysis for significant SNPs" (SNPs don't express), then they mention testing expression against a "group that includes the other 55,779 genes", which is a number that appears too high for the human genome (maybe use "gene expression" and "transcripts" instead?).

We thank the reviewer for pointing these out, and have revised these statements in line 535 and lines 448-450. We have changed "the other 55,779 genes expressed in CNCCs" into "the other 55,779 gene expression values reported in CNCCs".

And:

"We computed a genomic relationship matrix (GRM) combining genotype data for of all individuals for which data for at least one trait was available. The GRM was calculated using LDK5²⁵ with default parameters."

Reviewer #3 (Remarks to the Author):

This study presents a GWAS for facial variation in the CANDELA cohort, where the authors found a total of 42 genomic regions associated with facial features from 2D photographs. Overall, the manuscript is well written and I have the following comments to further clarify the study:

1. There is a lot of data in the study and it can be difficult to follow the analyses at times. It is unclear how many GWASs were conducted. Additionally,

summaries of these analyses, such as Manhattan plots and QQ plots should be included.

We agree with the reviewer that the number of conducted GWASs is not sufficiently clear. We have now specified in the Methods section the number of GWASs conducted in line 458. We have also uploaded all the Manhattan plots and QQ plots for each trait to Figshare, and provide a DOI number in the Methods section.

“Genome-wide association study (GWAS) was conducted on the 301 ILD phenotypes using PLINK v1.9. Sex, age, BMI, three head angles (yaw, pitch, roll) and the first 6 genetic PCs were included as covariates.”

2. It is unclear how the results from this analysis differ from the results from the analysis that used manual landmarks (or other studies of facial variation). For the results that were nominal significant in this analysis, what were their p-values in the previous analysis? This should be added to the study.

Following the reviewer’s suggestion, we now compare the results of this study with results from previous GWASes conducted in the CANDELA cohort for: (i) categorical traits, (ii) quantitative traits obtained from 3D images (on a subset of samples), and (iii) measurements derived from manual landmarks on 2D profile photographs. To illustrate these comparisons, we now provide scatterplots in Supplementary Figure S4, three additional columns in Supplementary Table S4, and mentioned this in the Results and Discussions section in lines 179-183.

“Among these regions, nine have been previously reported in other GWASs of facial features (including six that were detected in the previous GWASs of facial morphology we conducted in the CANDELA cohort; Supplementary Figure S4 and Supplementary Table S4).”

3. Pg 5: It is unclear how European/Native American ancestry is defined here. More details are needed

We thank the reviewer for pointing out this omission. We now add an explanation of how European/Native American ancestry is defined in lines 153-155, and provide details in the Methods section (lines 453-457) on how this was calculated.

“On the basis of genome-wide SNP data, we estimated continental ancestry proportions (European, Native American, and sub-Saharan African) for each CANDELA individual.”

“An LD-pruned set of 93,328 autosomal SNPs was used to estimate European, African and Native American ancestry proportions using supervised runs of ADMIXTURE⁶¹. Reference parental populations included in the ADMIXTURE analyses consisted of Africans and Europeans from 1000 Genomes Phase 3 and selected Native Americans, as described in Chacon-Duque et al (2018)⁶².”

4. Pg 6: It is unclear how both a p-value threshold and an FDR threshold were applied. Additionally, the exact threshold for the FDR cut-point is undefined. More details are needed

We apologize that our initial text was confusing because adjustment for multiple testing via FDR was being conducted in two completely

different situations, but this was not properly described, nor the two FDR procedures in the two situations distinguished. This issue has been pointed out by reviewer 2 as well, and we have made changes in the text to clarify this. In this instance, we have tried to clarify this as follows (in lines 172-175):

“We considered a P-value $< 5 \times 10^{-8}$ as threshold for significance, as this is stricter than the False Discovery Rate (FDR) multiple testing correction procedure of Benjamini-Hochberg (which results in a threshold of 1.82×10^{-6} , across SNPs and traits, see Methods).”

5. Pg 9: It is again unclear what the P-value and FDR threshold is and how these were applied.

As explained in response to the point above, this was the second situation in which FDR was applied to adjust for multiple testing. We have tried to clarify this as follows (in lines 197-199):

“We calculated a significance threshold for replication of 0.029 using Benjamini-Hochberg’s FDR procedure (accounting for 27 regions tested in four replication datasets).”

6. Pg 9: It is unclear how the candidate genes were selected for this analysis. More details are needed

We now provide more information of how the candidate genes were selected in Method in lines 582-588. We also changed Column T from “nearby genes” to “candidate genes” in Supplementary Table S4, and narrowed down the candidate genes presented in Table 2.

“Candidate genes were selected after checking their functions in Online Mendelian Inheritance in Man (OMIM: <https://omim.org/>) and Genecards (<https://www.genecards.org>) of nearby genes (within a 300 Kb window). Particularly, genes reported to be involved in craniofacial development, or mutations of which reported to cause facial dysmorphism in either humans or animal models were selected as candidate genes. When no such evidence was available, the nearest genes were chosen as candidates.”

7. Pg 9: It is unclear how the archaic introgression analysis was conducted, how these regions were defined, and how they were chunked into 140 segments. More details are needed.

We have tried to clarify this section by rephrasing the main paragraph and we also provide more information on these analyses in the Methods section in lines 557-573. In brief, we specify how we came to apply an admixture mapping approach to this specific situation and how we implemented it, resulting in the test of 103 segments. We hope that the readability of this section is now enhanced and satisfactory.

“In a previous study¹, we tested the association between an archaic introgression tract and various phenotypes by regressing these phenotypes on the number of copies of the introgression tract carried by each individual. This approach was then possible because the introgression was not variable in length or position, which is not longer the case for the ATF3 region under study here. We have therefore applied an admixture mapping approach to test the association between

archaic tracts and traits associated to that region. In this approach, we have first counted the number of alleles that were carried by an introgression tract for each imputed SNP in the region and for each individual. Genotypes at these SNPs were therefore coded as number of alleles located in an archaic tract. As archaic tracts span over tens of Kb (tracts with length <10Kb were not retained – see above), it is expected that all individuals (or almost all of them) have the same genotype at consecutive SNPs. We have therefore merged consecutive SNPs with a very similar distribution of genotypes throughout individuals, allowing a maximum of 1% genotype change over all individuals from a SNP to the next one. Eventually, we retained only the segments whose frequency was greater than 1%, that is, SNPs in that segment carry an archaic allele with a frequency >1%. This led to a total of 103 segments, which were tested for association using the same linear model as for the initial GWAS.”

8. Pg10: It is unclear what multivariate mixed model was used. More details are needed

We now explain further the approach used to reanalyse the mouse data in lines 204-224.

“To evaluate the potential effect in the mouse of the novel facial feature regions detected here, we reanalyzed genome-wide SNP data for 692 outbred mice previously characterized for craniofacial shape variation³¹. We applied a powerful multivariate mixed model not used in the original analysis of these data³¹. The original mouse GWAS was done on each shape PC³¹. However, this approach has maximum power when an allele effect is strong enough to sufficiently structure the overall shape variation. In this case, a phenotypic PC could become collinear with this genetic effect. With geometric morphometrics on skull shape, this is really unlikely and a multivariate GWAS is preferred. It adequately models a complete pleiotropy among Procrustes coordinates. Such an approach is nevertheless computationally challenging when a linear mixed model (mvLMM) based on the genomic relatedness matrix is used on a very high dimensional trait such as skull shape (here 67 non-null dimensions). We therefore approximated this mvLMM by modeling the covariance matrices of this linear mixed model with two blocks. The first block models the genetic and environmental covariances of the first 10 PCs (62% of the total shape variance) altogether, while only the variances for the next 57 traits were modelled as the second block (i.e. the covariances among these PCs as well as with the other block were set to 0). We then used the Pillai trace statistics to assess the effect of each SNP. In comparison to the original mapping³¹, this approach gains from the modelling of the genetic correlations between the main PCs while maintaining a lower dimensionality cost than in the full multivariate model. Moreover, we follow the procedure of Nicod et al. (2016)³² by deriving false discovery rate based on 100 permutations to identify SNPs exceeding a FDR threshold of 5%.”

9. Pg 10: It is unclear what the strong association was between. More details are needed

To clarify, we now state that (in lines 226-229):

“A region on mouse chromosome 5 (homologous to human 22q12.1), showed strong association between allele dosage at the SNP and skull shape (maximum P-value: 2×10^{-34}), far exceeding a multiple testing-adjusted significance threshold.”

10. Pg 11: Besides a brief mention that the manual landmarking and the landmarks provided by Face++ were correlated, there was no evidence presented that this software is reliable and this statement should be removed from the conclusion.

Taking on board the reviewer’s suggestion, we have expanded the text in the Methods and Results sections of the manuscript to further elaborate on the reliability of Face++. It is important to compare to the results of the manual landmarking, because the landmarking performed by an expert rater is taken to be the ground truth to which the automated landmarking results are compared. In addition, to further evaluate the reliability of the Face++ landmarking, we now examine another commonly used automatic landmarking tool, Dlib[5]. We provide a comparison between Face++ and Dlib, and with manual landmarking. The comparison results are included in Supplementary Table S1. We also incorporate Supplementary Figure S2 in the submitted version with Supplementary Table S1. A brief description of this result is also added in the Results section in lines 433-438. The consistency of Face++ landmarks compared to manual landmarks were similar to, and for some landmarks better than the commonly used tool Dlib, according to both ICC and pixel distance, which indicates the reliability of Face++ landmarking. Besides, Face++ provided more landmarks (106) than Dlib (68), especially in anatomically important regions such as nasal bridge. Other than the comparison of landmarking reliability across methods, we believe that the replication of previous facial association results supports the reliability of Face++ landmarking. We have now qualified the statement made in the conclusion.

11. Pg13: It is unclear if there was any additional quality control for imputation probability or MAF was conducted after imputation. This should be added to the methods.

We thank the reviewer for pointing out this omission. We now provide more details concerning the quality control checks for imputation probability or MAF in the Methods section in lines 395-402.

“Markers that are monomorphic in 1000 Genomes Latin American samples were excluded from imputation. Chip genotyped SNPs having a low concordance value (< 0.7) or a large gap between info and concordance values ($info_type0 - concord_type0 > 0.1$), which might be indicators of poor genotyping, were also removed, both from the imputed and chip dataset. Imputed SNPs with imputation quality scores < 0.4 were excluded. The IMPUTE2 genotype probabilities at each locus were converted into best-guess genotypes using PLINK (at the default setting of < 0.1 uncertainty). SNPs with $> 5\%$ uncalled genotypes or minor allele frequency $< 1\%$ were excluded.”

12. Pg 13: The statement on multiple test correction is unclear was a FDR threshold used or was a Bonferroni correction? What analysis was this threshold used for?

We acknowledge that the statement we made on multiple test correction was not very clear. We now provide more details and rephrased this section in lines 481-498. The text was included in the response to Reviewer #2.

13. Pg 14: the statement on FDR and combined significance is unclear and it is not apparent what analysis this is for.

We have tried to clarify this by providing more details and specifying which analysis we refer to, as follows (in lines 513-515):

“In the replication analysis, 76 p-values were available for 27 novel associated regions across 4 separate datasets. After correction of multiple testing with BH-FDR, the combined significance threshold in the replication cohorts was 0.0293.”

Reference

1. Cole, J.B., et al., *Genomewide Association Study of African Children Identifies Association of SCHIP1 and PDE8A with Facial Size and Shape*. Plos Genetics, 2016. **12**(8).
2. Null, M., et al., *Genome-wide analysis of copy number variants and normal facial variation in a large cohort of Bantu Africans*. Human Genetics and Genomics Advances, 2022. **3**(1).
3. Conomos, M.P., et al., *Model-free Estimation of Recent Genetic Relatedness*. American Journal of Human Genetics, 2016. **98**(1): p. 127-148.
4. Xiong, Z.Y., et al., *Novel genetic loci affecting facial shape variation in humans*. Elife, 2019. **8**.
5. Sagonas, C., et al., *300 Faces In-The-Wild Challenge: database and results*. Image and Vision Computing, 2016. **47**: p. 3-18.

Reviewers' comments:

Reviewer #1 (Remarks to the Author):

Li and colleagues have carefully responded to my comments. Their explanations are reasonable, and details provided are sufficient. I appreciate that the authors expanded the datasets included for comparison analyses. I believe that the study is improved with these new analyses and helped to strengthen this manuscript. I don't have any further comments or recommendations.

Reviewer #2 (Remarks to the Author):

Li et al have submitted an improved version of their manuscript "Fully automatic landmarking of 2D photographs identifies novel genetic loci influencing facial features". In this new submission, several instances of lack of clarity were clarified and the authors have generally shown a willingness to address points that were of concerns to the reviewers.

There are a few issues that (unfortunately remain) in the manuscript, despite the undeniable efforts made by the authors. The main issue in the opinion of this reviewer is how the authors dealt with population structure and admixture. The authors: argue that analyzing all samples using a PC-based approach combined with a linear mixed model is sufficient to account for admixture and the p-values they reported are similar to the ones using "GENESIS" (some software aimed at calculating population and pedigree structure). The evidence they provide is a previous Plos One paper and their own publication. There is much to disagree with this approach. The use of principal component and similar methods to correct for admixture is far from being widely accepted. There are several reasons why global-adjustment methods may be considered unreliable and previous work has shown that, among others, "incorporating principal components in linear mixed models may not be adequate and indeed may yield a substantially similar estimate as the non-adjusted model" (Derks et al Behav Genet, 47(3), pp.360-368). Population admixture has traditionally been difficult to deal with and the analytical tools have evolved and become increasingly sophisticated (Atkinson, Nat genet, 53(2), 195-204.). Although this reviewer has much sympathy, the authors' argument, that they ran analyses that must be valid because they produced results that were consistent with those produced by better methods (which, as commented above, they were not), equates to obtaining putatively correct results using suboptimal methodologies. The authors and the reviewers alike have an obligation to promote the best practice in science.

A few other secondary issues also remain. The authors report Spearman's correlations, but they still use "r" (instead of the Greek "rho" or "rs"). But the initial question was not so much on the use of Pearson's vs Spearman's correlation, but rather whether correlations with binary variables are interpretable. In this case, there will be a lot of rank sharing since the biological sex in these samples was a binary variable. This is mathematically similar to a rank biserial analysis and a Mann-Whitney test. Conceptually, would a true comparison of the means be the most straightforward analysis to conduct?

The text still reads "Expression analysis for significant single nucleotide polymorphisms (SNPs)" –SNPs don't express, but genes annotated to them may. And that "as well as for all significantly associated SNPs with and $r \leq 0.1$ with the index SNP" – why were conditional analyses, mentioned previously in the text (line 509), results not used ?

And "55,779 gene expression values"; it may be that the authors refer to a list that includes multiple transcripts of the same genes, or a lot of pseudogenes, since the total number of genes is around 20,000.

Reviewer #3 (Remarks to the Author):

This is a revision of a manuscript that presents a GWAS for facial variation in the CANDELA cohort, where the authors associated genomic regions with facial features from 2D photographs. Since the last version of the manuscript, there have been changes to clarify the methods and results described in the manuscript, however several things are still unclear or missing, therefore, I have the following comments:

- 1) Line 156 – “themselves strongly negatively correlated” – unclear what this is referring to.
- 2) Line 189-195 – awkwardly phrased and unclear if all 3 data sets were processed similar to the CANDELA study or if just the first replication dataset was.
- 3) Line 198 – why was a Bonferroni correction not done here similar to the other analyses presented in the paper?
- 4) Line 202 – There is missing data/blank lines in Table S4 and it’s unclear why
- 5) Table S4 - Why is there no correlation between results from the previous paper and this one when the landmarks themselves were so highly correlated?
- 6) Line 207-224 – this description of the LMM used for the mouse data is out of place in the results section and should be moved to the methods
- 7) Line 226 – The full results for these 30 regions should be included in the manuscript
- 8) Line 238-285 – this should be condensed/summarized and mostly moved to the discussion.
- 9) Line 243-258 – The archaic human ingression analysis is still unclear, both in how it was done and what it means. It is also only discussed for this one region and not discussed at all in the discussion. Why wasn’t this analysis done for the other regions associated in this study?
- 10) Line 287-289- unclear If these annotation are for the MANE transcript or if one transcript was somehow selected.
- 11) Line 298-300 – Are these the only two regions that overlap with craniofacial enhancers/promoters? If so the word “often” should be removed.
- 12) Line 300-304 – unclear how candidate genes were selected and how enrichment tests were done. This information is also not in the methods section
- 13) Discussion – there is little discussion of the scientific, biological results from this study. This should be expanded upon. A majority of the discussion revolved around the use of 2D photographs which is not the main focus of the results and therefore it is confusing as to why it is main focus of the discussion
- 14) Line 443-445: It is unclear what these sentences mean. How were the phenotypic characteristics explored and estimated using a correlation coefficient? What were the correlation coefficients? Were these values not provided by participants upon recruitment?
- 15) Cannot access the Manhattan plots and QQ plots – The web address provided does not work and these should be added to the supplement
- 16) Line 483-500: It isn’t clear how you would get an adjusted p-value of 1.823×10^{-6} from your number of effective tests, but I also don’t think it is necessary for to include in the manuscript since a 5×10^{-8} threshold is ultimately used.
- 17) Line 509-514: Regions should be reported previously seen if they’ve been reported in the literature. Follow up can then be done to see if there are multiple signals at the site. Additionally, it is unclear for the conditional analyses how reported SNPs that were not in the current dataset was handled and how many of these analyses were done/what SNPs were included. Results from conditional analyses should be included.
- 18) Line 515-517 – unclear again how the 0.0293 was calculated, but more importantly why is a more stringent Bonferroni for 0.05/27 ($p < 0.001$) not used here since this would be consisted with the approach used in the GWASes above?
- 19) Line 518 – this interaction analysis was never discussed in the results. Additionally, unclear what all interactions were tested since there are 52 interaction terms but only 42 regions discussed

elsewhere in the paper. What regions were tested?

20) Line 531 – unclear what species these CNCCs are from or anything about the original study. Also unclear how genes were assigned to the regions. Was it distance from the index SNP or some other threshold measure? Was any QC done on the RNA-seq data prior to it being used in this study?

21) Table 2 – why is the data for only 5 regions presented here? I thought there were 33 newly associated regions?

22) Line 584 – Were nearby genes selected based on the OMIM information and then put into the GO enrichment test? If so, it should be redone with just a nearby gene criteria so as to not bias the test.

Minor:

Line 187 – and should be “an”

RESPONSE THE REVIEWERS' COMMENTS.

Our replies below are shown in **bold**. Related changes have been highlighted in the manuscript file (quotes taken from the manuscript are shown here in *italic*).

In addition to the specific changes made in response to the reviewers' comments, we have:

1. Added two authors (M.E. Delgado, M.R. Khokan), who performed the analysis on Neandertal facial features section that we included in this revised manuscript.
2. Merged Figure 2 and Figure 3, and updated the locus zoom plot for 22q12.1, to include a wider range matching the locus zoom plot for mouse.
3. Broken down the previous "Results and Discussion" section into separate sections for "Results" and "Discussion".

Reviewers' comments:

Reviewer #1 (Remarks to the Author):

Li and colleagues have carefully responded to my comments. Their explanations are reasonable, and details provided are sufficient. I appreciate that the authors expanded the datasets included for comparison analyses. I believe that the study is improved with these new analyses and helped to strengthen this manuscript. I don't have any further comments or recommendations.

Reviewer #2 (Remarks to the Author):

Li et al have submitted an improved version of their manuscript "Fully automatic landmarking of 2D photographs identifies novel genetic loci influencing facial features". In this new submission, several instances of lack of clarity were clarified and the authors have generally shown a willingness to address points that were of concerns to the reviewers.

1. There are a few issues that (unfortunately remain) in the manuscript, despite the undeniable efforts made by the authors. The main issue in the opinion of this reviewer is how the authors dealt with population structure and admixture. The authors: argue that analyzing all samples using a PC-based approach combined with a linear mixed model is sufficient to account for admixture and the p-values they reported are similar to the ones using "GENESIS" (some software aimed at calculating population and pedigree structure). The evidence

they provide is a previous Plos One paper and their own publication. There is much to disagree with this approach. The use of principal component and similar methods to correct for admixture is far from being widely accepted. There are several reasons why global-adjustment methods may be considered unreliable and previous work has shown that, among others, “incorporating principal components in linear mixed models may not be adequate and indeed may yield a substantially similar estimate as the non-adjusted model” (Derks et al Behav Genet, 47(3), pp.360-368). Population admixture has traditionally been difficult to deal with and the analytical tools have evolved and become increasingly sophisticated (Atkinson, Nat genet, 53(2), 195-204.). Although this reviewer has much sympathy, the authors’ argument, that they ran analyses that must be valid because they produced results that were consistent with those produced by better methods (which, as commented above, they were not), equates to obtaining putatively correct results using suboptimal methodologies. The authors and the reviewers alike have an obligation to promote the best practice in science.

We fully agree with the reviewer in that GWAS in admixed populations present opportunities as well as challenges. The issue of correcting for population stratification (globally or locally along the genome) is an important issue to consider. We were as excited as others working on admixed populations with the publication of Atkinson et al (Nat. Genet. 53,195–204, 2021) proposing TRACTOR. Particularly the fact that this approach accounts for local ancestry along the genome and ancestry-specific allelic effects. We had therefore explored TRACTOR, since it was published, included having various exchanges with Dr Atkinson regarding its implementation and our experience with it. However, the results we have obtained have been disappointing, and this appears to be the experience for others. This has been recently discussed in a paper by Hou et al (Nat Genet. 2021 53: 1631–1633), who summarized some important limitations of the TRACTOR approach. Two issues highlighted by Hou et al., which are of special relevance here, are that: (i) Tractor is under-powered compared to traditional GWAS for admixed populations, unless there is allelic effect size heterogeneity by ancestry, and (ii) traditional GWAS methods, which correct for global rather than local

ancestry, do not suffer from inflation (this last point relates to another comment made by this reviewer). Our experience with TRACTOR is in line with these two criticisms of Hou et al. 2021. Furthermore, we have observed that in cases in which there is extreme allelic differentiation between the parental populations contributing to the admixture, TRACTOR has an almost total loss of power. This occurs for instance for the *EDAR* missense SNP rs3827760 (which has been very robustly associated with facial features in many studies, and also validated by mouse experiments). The derived allele at rs3827760 is entirely absent in European and African ancestries, but is nearly fixed in Native Americans (frequency of 98%). In this case, the Native American local ancestry component is nearly identical to the SNP genotype, and therefore TRACTOR suffers from high collinearity and loses power completely.

To incorporate the reviewer's concerns to the revised manuscript we added in the M&M a section devoted to the issue of correction for population stratification (section entitled: "*Genome-wide association analyses and correction for population stratification*"). This section is accompanied by the additional Supplementary Figures S8-S10 and Supplementary Table S8). In preparing this new section, we fully re-analysed the data using four different approaches, in order to compare them with the main PLINK results: (i) GENESIS which includes both genetic PCs and a kinship matrix; (ii) GCTA, which includes a Genetic Relatedness Matrix (GRM, a specific request made by the reviewer), (iii) TRACTOR, and (iv) SNP1 (another local ancestry based method), which was reported by Hou et al. as more powerful than TRACTOR. In the novel M&M section and in Supplementary Figures S8-S10 and Supplementary Table S8 we describe fully the results obtained. In summary, the main findings are that:

- There is substantial inflation in absence of any correction for population structure, which entirely goes away with any form of whole-genome based adjustment (PLINK or GCTA or GENESIS). We show that the results of these three methods are nearly identical. We also provide some additional citations from both simulation studies and other studies on admixed populations, including Latin Americans,

which report similar observations.

- While global ancestry (either as PCs or GRM or both) is sufficient by itself to correct for stratification, local ancestry alone is not sufficient. Local ancestry based methods (TRACTOR and SNP1) need to also include global ancestry (as genetic PCs) to fully correct for stratification. This was also the suggestion that we received directly from Dr Atkinson.
- Consistent with the findings of Hou et al, we find that for well-validated face genes the global ancestry correction models have highest power, followed by SNP1, while TRACTOR has the lowest power. This is including (but not limited) to regions like *EDAR* where SNPs are almost fully indicative of local ancestry. This low power of TRACTOR in our case could stem from the effect size not being sufficiently different between ancestral components to reach genome-wide significance in each ancestral component. TRACTOR is unique, compared to the other approaches, in that it has three degrees of freedom with three ancestry-specific SNP components. This means that (as concurred by Atkinson et al. 2021 in their reply to Hou et al.), TRACTOR is by design more powerful than other tests only when there is substantial heterogeneity in effect size of the SNP across ancestry backgrounds. The trade-off between the scarcity of variants with ancestry-specific effect sizes and that of variants with effect size shared across ancestries would therefore be a handicap of the TRACTOR approach.

In conclusion, the analyses we have carried out lead us to believe that in our case: (i) the genetic PC correction used in PLINK performs at least as well as other similar approaches proposed to date and (ii) local-ancestry correction approaches, such as that of Atkinson et. al, are underpowered.

2. A few other secondary issues also remain. The authors report Spearman's correlations, but they still use "r" (instead of the Greek "rho" or "rs"). But the initial question was not so much on the use of Pearson's vs Spearman's correlation, but rather whether correlations with binary variables are interpretable. In this case, there will be a lot of rank sharing since the biological sex in these samples was a binary variable. This is mathematically similar to a rank biserial analysis and a Mann-Whitney test. Conceptually, would a true

comparison of the means be the most straightforward analysis to conduct?

We thank the reviewer for pointing this out. We now have changed Spearman's r to Greek ρ in the manuscript. In addition, for binary variables we now report the point biserial correlation coefficient, and use r_{pb} to refer to the point biserial correlation coefficient (between sex and phenotypes). The relevant values have also been updated in Supplementary Table S3. The reviewer is correct that we should have gone for a test comparing of the means if statistical significance was our objective. But instead, the magnitude and direction of the correlation are of our primary interest, to understand the nature of relationship between the variables, which is why we have reported the correlations in our manuscript.

3. The text still reads "Expression analysis for significant single nucleotide polymorphisms (SNPs)" –SNPs don't express, but genes annotated to them may. And that "as well as for all significantly associated SNPs with and $r \leq 0.1$ with the index SNP" – why were conditional analyses, mentioned previously in the text (line 509), results not used?

The reviewer is correct and apologise for the incorrect phrasing used.

We now have changed "Expression analysis for significant single nucleotide polymorphisms (SNPs)" to: "Enrichment analysis and comparison of expression levels for nearest candidate gene".

And the section including the sentence "as well as for all significantly associated SNPs with and $r \leq 0.1$ with the index SNP" has been fully rephrased so as to clarify this sentence and also avoid the confusion between the replication analyses described in this section and the conditional association analyses described elsewhere.

"When data for the index SNP of a region identified in the CANDELA sample was not available in the other study samples, we examined as proxies SNPs in LD with the index SNP in a region ($r^2 \geq 0.1$)."

4. And "55,779 gene expression values"; it may be that the authors refer to a list that includes multiple transcripts of the same genes, or a lot of pseudogenes, since the total number of genes is around 20,000.

The reviewer is correct in that the number transcripts is much higher than the number of individual protein genes. This is due to the fact that, other than multiple transcripts per protein coding gene, the RNAseq values used also include non-protein coding transcripts.

Reviewer #3 (Remarks to the Author):

This is a revision of a manuscript that presents a GWAS for facial variation in the CANDELA cohort, where the authors associated genomic regions with facial features from 2D photographs. Since the last version of the manuscript, there have been changes to clarify the methods and results described in the manuscript, however several things are still unclear or missing, therefore, I have the following comments:

1. Line 156 – “themselves strongly negatively correlated” – unclear what this is referring to.

We apologized for the confusion. We meant that European and Native American ancestry proportions are strongly negatively correlated. We have now rephrased this in lines 406-409 as follows:

“On the basis of genome-wide SNP data, we estimated European, Native American and sub-Saharan African ancestry proportions for each CANDELA individual (European and Native American ancestries being strongly negatively correlated).”

2. Line 189-195 – awkwardly phrased and unclear if all 3 data sets were processed similar to the CANDELA study or if just the first replication dataset was.

We understand and have rewritten this section in lines 183-190 as follows:

“Considering the admixed ancestry of the CANDELA sample, we sought replication in study samples with different continental ancestries, East Asian, European and African. For East Asians, we had available frontal 2D photographs and genome-wide SNP data for 5,078 individuals^{31, 32}. These data were therefore processed as for the CANDELA sample. For Europeans and Africans, we extracted association P-values from published studies: a GWAS meta-analysis including data for 10,115 Europeans and 78 inter-landmark distances¹⁷ and a GWAS performed in

3,631 African individuals for 34 size and shape-related facial traits (distances and Principal Components)^{5, 33}.

3. Line 198 – why was a Bonferroni correction not done here similar to the other analyses presented in the paper?

We thank the reviewer for pointing out that in some cases False Discovery Rate and in some other cases Bonferroni method was used to correct for multiple testing. We have now revised the manuscript to use False Discovery Rate to correct for multiple testing in all follow-up analyses throughout the manuscript. We note that, as the follow-up analyses are applied to candidates that were detected via some prior analyses, the FDR method is the more appropriate method in such cases, and more powerful too, while the Bonferroni method can often be overly conservative (Nakagawa 2004, Behavioral Ecology, among others).

4. Line 202 – There is missing data/blank lines in Table S4 and it's unclear why

The blank lines refer to the nine regions from previous studies that we replicate here. Since these are established loci (i.e. not novel) we did not seek their replication in other datasets. That is, we only sought replication in independent datasets for the 33 novel loci we identified in the CANDELA data. We now indicate this in the header of Supplementary Table S4 as: “Replication P-values for novel regions”.

5. Table S4 - Why is there no correlation between results from the previous paper and this one when the landmarks themselves were so highly correlated?

We do find that several loci replicate across studies, including 6 of the 9 shown in Table 1 of the manuscript. There are however important differences between our three studies. The first one (Nat. Comms doi: 10.1038/ncomms11616) was based mostly on categorical phenotyping of selected facial traits. A limited amount of landmarking was done in order to confirm the categorical findings. This landmarking was limited to 2,955 CANDELA individuals and 34 landmarks (of which only 16 match those placed here by Face ++). In addition these landmarks were used to obtain distance measures related to the specific phenotypes, therefore not matching the distances obtained here. Our second GWAS (Science Advance 10.1126/sciadv.abc6160) was based on landmarking of lateral

(profile) photos. Therefore few landmarks match the ones placed by Face++. Furthermore the distances analysed in that study are mainly sensitive to (anterio-posterior) face protrusion, a dimension to which the distances analysed in the current study are not very sensitive. Considering this question from the reviewer, we expanded the description of these differences between studies in the revised manuscript.

6. Line 207-224 – this description of the LMM used for the mouse data is out of place in the results section and should be moved to the methods

Following the reviewer's suggestion, we have now moved this to the Methods section.

7. Line 226 – The full results for these 30 regions should be included in the manuscript

We now provide information on the 30 human regions mapped to the mouse genome in Supplementary Table S5.

8. Line 238-285 – this should be condensed/summarized and mostly moved to the discussion.

We have reworked and condensed this section, and moved most of the previous text to the new Discussion section.

9. Line 243-258 – The archaic human ingression analysis is still unclear, both in how it was done and what it means. It is also only discussed for this one region and not discussed at all in the discussion. Why wasn't this analysis done for the other regions associated in this study?

We thank the reviewer for this comment. It motivated us to fully rework the presentation of our introgression results (including performing additional analyses) and to highlight this work throughout the paper (including mentioning it in the title). The presentation of the methodology has been reworked (in lines 634-640) and the results are now presented in a separate section (in lines 233-260).

Of special note, in performing these revisions we accessed available nasal height measurements for Neanderthal and modern human crania, which correspond to a measurement obtained in the CANDELA volunteers. This allowed us to assess the direction of the effect we

estimated in the CANDELA data. We find that this effect is consistent with the Neanderthal-modern human differentiation measured in crania. Namely, Neanderthals have greater nasal height than modern humans and we see that in the CANDELA individuals, Neanderthal introgression increases nasal height. To our knowledge this is the first time that Neanderthal introgression is shown to have an effect consistent with the morphological differentiation between Neanderthals and modern humans, and consequently we now highlight this result in the paper.

In response to the reviewer's last question: we focused on 1q32.3 due to the fact that archaic introgression in this region had been reported previously and we wanted to evaluate the relationship between our GWAS signal and introgression in this region (Chintalapati, M., M. Dannemann, and K. Prufer, *Using the Neanderthal genome to study the evolution of small insertions and deletions in modern humans*. BMC Evolutionary Biology, 2017. 17.; Sankararaman, S., et al., *The Combined Landscape of Denisovan and Neanderthal Ancestry in Present-Day Humans*. Curr Biol, 2016. 26(9): p. 1241-7). Following the reviewer's suggestion, we performed the introgression analyses for all the other novel regions detected in the CANDELA sample and confirmed that the only clear signal of introgression is in 1q32.3.

10. Line 287-289- unclear If these annotation are for the MANE transcript or if one transcript was somehow selected.

We apologize for the lack of clarity. In fact, FUMA (the tool we used to do annotation) uses ANNOVAR to annotate the SNPs. Their website indicates that it uses all annotated transcripts in the Gencode collection lifted up to hg19, and returns the annotation value based on a prioritization table. Details are now provided in lines 661-664.

“The website indicates that ANNOVAR uses all annotated transcripts in Gencode collection lifted up to hg19, and has its own prioritization criteria to report the most deleterious function. Only prioritized annotations are used for those SNPs.”

11. Line 298-300 – Are these the only two regions that overlap with craniofacial enhancers/promoters? If so the word “often” should be removed.

These are two of many that are either near or within craniofacial enhancers/promoters. These are now indicated in Supplementary Table S4. We also have removed previous Supplementary Figure S6 and incorporated this information into Supplementary Table S4.

12. Line 300-304 – unclear how candidate genes were selected and how enrichment tests were done. This information is also not in the methods section

We apologize for the lack of clarity regarding these analyses. Following by the reviewer's suggestion (point #22), we now use the 43 genes nearest to the index SNP in each region (see Supplementary Table S4) as input for GO enrichment analysis done with Metascape. We now add this description in the Methods, lines 666-668.

"We used Metascape (<http://metascape.org/>) to carry out a GO analysis of genes nearest to the index SNPs of the novel associated regions (if an index SNP was in two genes, both genes were retained in the analysis) (Supplementary Table S4)."

13. Discussion – there is little discussion of the scientific, biological results from this study. This should be expanded upon. A majority of the discussion revolved around the use of 2D photographs which is not the main focus of the results and therefore it is confusing as to why it is main focus of the discussion

Following the reviewer's suggestion, we have now split our previous Results/Discussion section into separate, individual, sections. The new Discussion section has been rewritten and expanded, with greater emphasis given to the biological significance of our results (highlighting the introgression findings) and reducing our comments on the 2D methodology.

14. Line 443-445: It is unclear what these sentences mean. How were the phenotypic characteristics explored and estimated using a correlation coefficient? What were the correlation coefficients? Were these values not provided by participants upon recruitment?

We apologize for these confusing sentences. The text is now as follows:

"Correlations between inter-landmark distances and covariates (gender, age, BMI, genetic ancestry and head angles) were evaluated using point

biserial correlation coefficient (with gender) and Spearman's correlation coefficient (with age, BMI, genetic ancestry and head angles)."

The reviewer is correct that information on gender, age and BMI was obtained upon recruitment. Values for the other covariates were obtained here. All the calculated correlation coefficients (and their statistical significance) are presented in Supplementary Table S3. The main observations from this table are described and related comments made in the text.

15. Cannot access the Manhattan plots and QQ plots – The web address provided does not work and these should be added to the supplement

We apologize for the web address provided not working. We have contacted the figureshare website to fix this. Now the web address provided (<https://doi.org/10.6084/m9.figshare.19728916>) is working.

16. Line 483-500: It isn't clear how you would get an adjusted p-value of 1.823×10^{-6} from your number of effective tests, but I also don't think it is necessary for to include in the manuscript since a 5×10^{-8} threshold is ultimately used.

The reviewer is correct that we ultimately do not use the genomewide FDR threshold for GWAS significance, using the classical threshold instead. But the genomewide FDR threshold was calculated as per the request of reviewer 1 in the first round of revisions, who suggested that we employ some other methods of estimating a multiple testing correction factor for comparisons. This seems to have satisfied the reviewer, who has kindly indicated in their response in this round of review that our implemented changes were satisfactory.

17. Line 509-514: Regions should be reported previously seen if they've been reported in the literature. Follow up can then be done to see if there are multiple signals at the site. Additionally, it is unclear for the conditional analyses how reported SNPs that were not in the current dataset was handled and how many of these analyses were done/what SNPs were included. Results from conditional analyses should be included.

Amongst 42 detected regions, 16 are entirely new (no SNP has been associated previously in these regions). We conducted the conditional analyses for the other 26 regions. In total we tested 93 SNPs across 26

regions in the conditional analyses (the details now can be seen in Supplementary Table S6).

We have clarified this in the text in lines 499-508:

“Conditional GWAS was also carried out to test if a signal detected here had been reported previously. We firstly picked out signals that fall on the chromosome bands that have been reported. Amongst 42 regions we detected, 16 fell on an entirely new chromosome band that was not reported to be associated with facial features. We then have conducted the conditional analysis on totally 93 SNPs across 26 regions. We gathered all reported SNPs in each chromosome band and added those reported SNPs into the regression models of corresponding SNPs of the same chromosome band in our results. If P-value obtained was above the suggestive significant threshold (1×10^{-5}), this signal would be regarded as a reported signal, and conversely, it would be regarded as a new signal. Details of the results from conditional analyses could be seen in Supplementary Table S7.”

18. Line 515-517 – unclear again how the 0.0293 was calculated, but more importantly why is a more stringent Bonferroni for $0.05/27$ ($p < 0.001$) not used here since this would be consistent with the approach used in the GWASes above?

As we mention in response to the reviewer’s point 3 above, the reviewer is correct in pointing out that in some cases False Discovery Rate and in some other cases Bonferroni method was used to correct for multiple testing. We have now revised the manuscript to use False Discovery Rate to correct for multiple testing in all follow-up analyses throughout the manuscript. We note that, as the follow-up analyses are applied to candidates that were detected via some prior analyses, the FDR method is the more appropriate method in such cases, and more powerful too, while the Bonferroni method can often be overly conservative (Nakagawa 2004, Behavioral Ecology, among others).

19. Line 518 – this interaction analysis was never discussed in the results. Additionally, unclear what all interactions were tested since there are 52 interaction terms but only 42 regions discussed elsewhere in the paper. What regions were tested?

We thank the reviewer for pointing out this omission. The interaction analysis was discussed in the Results section in our first submission of this manuscript. Following revision, we moved the discussion of our nine reported regions into the supplementary note, which included the section of interaction analysis with *EDAR*. We now have removed the interaction analysis section from the Methods.

To answer the question, there are more interaction terms than number of regions because some regions were associated with multiple traits and all associations were tested.

20. Line 531 – unclear what species these CNCCs are from or anything about the original study. Also unclear how genes were assigned to the regions. Was it distance from the index SNP or some other threshold measure? Was any QC done on the RNA-seq data prior to it being used in this study?

We are sorry for the lack of details. To clarify, we used the human CNCCs RNA-seq data from of Prescott et al. (2015). In addition, we used published data from ENCODE (ENCODE Project Consortium, 2012). We examined the nearest genes, based on the distance to the index SNP in an associated region.

Since the RNA-seq data from of Prescott et al. (2015) had been processed using variance-stabilizing transformation (VST), we applied VST (using DESeq2) to the downloaded files of count data of tissue cell and primary cell from ENCODE. We then normalized RNA-seq VST values of CNCC, tissue cell, primary cell, and all types (including tissue cell and primary cell) for the comparative analyses. We now have added this information in the Methods section.

“We used Metascape (<http://metascape.org/>) to carry out a GO analysis of genes nearest to the index SNPs of the novel associated regions (if an index SNP was in two genes, both genes were retained in the analysis) (Supplementary Table S4). To examine patterns of transcription in the vicinity of index SNPs for the novel regions identified here, we contrasted the CNCC RNA-seq data from the study of Prescott et al. (2015) to that obtained by the ENCODE45 project for 318 different cell types (Supplementary Table S9). Of the 33 newly associated regions,

overlapping transcripts in the CNCC RNAseq data have been reported for 26, and only these could therefore be tested. For consistency with the CNCC data, we applied variance-stabilizing transformation (VST) to the ENCODE data (using DESeq2). The higher transcription levels in CNCCs, relative to the ENCODE data, was tested using a Student's t test, with a Benjamini-Hochberg's FDR threshold ($p < 0.034$).

21. Table 2 – why is the data for only 5 regions presented here? I thought there were 33 newly associated regions?

The regions presented in Table 2 are the 5 regions with strongest association with facial traits. To avoid making the manuscript excessively long, these are the only regions we comment on in the main text. The remaining novel regions presented in Supplementary Table S4. We have clarified this in the text and the legend to Table 2.

22. Line 584 – Were nearby genes selected based on the OMIM information and then put into the GO enrichment test? If so, it should be redone with just a nearby gene criteria so as to not bias the test.

Following the reviewer's suggestion, we redid this test selecting the genes based only on their distance to the index SNP (and focusing on the closest gene). The results are still significant. We have updated Supplementary Figure S7 accordingly, as well as the gene list in Supplementary Table S4. We now describe nearby gene selection in the text as follows:

"We used Metascape (<http://metascape.org/>) to carry out a GO analysis of genes nearest to the index SNPs of the novel associated regions (if an index SNP was in two genes, both genes were retained in the analysis) (Supplementary Table S4)."

23. Minor: Line 187 – and should be "an"

We have corrected this.

Reviewers' comments:

Reviewer #2 (Remarks to the Author):

The authors have largely addressed reviewers' concerns and have shown willingness to conduct additional and substantial analyses to that end.

This reviewer feels that some issues were embedded in the original study design and that correction for PCs which is the primary method used in this manuscript is a suboptimal mitigating strategy and which, in the opinion of this reviewer, may end-up legitimizing a method that could seriously sway the results in many GWAS studies. (the authors focus on p-values and lambdas, in a relatively small sample).

The authors have made some truly impressive efforts to address that criticism; this reviewer would recommend that the authors add a remark in their discussion to caution against a blanket use of PCs in GWAS studies, for which serious doubts remain.

Reviewer #3 (Remarks to the Author):

This is a revision of a manuscript that presents a GWAS for facial variation in the CANDELA cohort, where the authors associated genomic regions with facial features from 2D photographs. Since the last version of the manuscript, there have been many changes to clarify the methods and results described in the manuscript, and my previous comments have been addressed. I only have the following minor edits:

Line 131: "was superior to" – not supported by Table S1, which seems to show that the two methods perform similarly.

Line 143: how many individuals were removed?

Table S3 : the p-value correlation coloring is not clear in this table.

Line 110: "one region also impacts on mouse craniofacial morphology" – awkward phrasing.

RESPONSE THE REVIEWERS' COMMENTS.

Our replies below are shown in **bold**. Related changes have been highlighted in the manuscript file (quotes taken from the manuscript are shown here in *italic*).

Reviewers' comments:

Reviewer #2 (Remarks to the Author):

1. The authors have largely addressed reviewers' concerns and have shown willingness to conduct additional and substantial analyses to that end.

This reviewer feels that some issues were embedded in the original study design and that correction for PCs which is the primary method used in this manuscript is a suboptimal mitigating strategy and which, in the opinion of this reviewer, may end-up legitimizing a method that could seriously sway the results in many GWAS studies. (the authors focus on p-values and lambdas, in a relatively small sample).

The authors have made some truly impressive efforts to address that criticism; this reviewer would recommend that the authors add a remark in their discussion to caution against a blanket use of PCs in GWAS studies, for which serious doubts remain.

We thank the reviewer for his/her appreciation of the work we have done in revising our manuscript. We agree that analyses in admixed populations present particular analytical challenges. Incidentally, another paper on this topic is now in preprint (<https://www.biorxiv.org/content/10.1101/2023.01.20.524946v1>). As requested by the reviewer, we now include a remark to this effect in the last paragraph of the discussion:

“Populations with admixed continental ancestry, such as Latin Americans, offer challenges and opportunities for such studies. In these populations, optimal correction for population stratification, considering both global and local genomic ancestry, is a challenging analytical problem for which an all-round solution is yet to be developed^{63,64}. Nevertheless, the extensive genetic and phenotypic diversity of Latin Americans is enabling GWASs that have led to important insights into the

genetics of physical appearance^{1,2,22-24}.”

Reviewer #3 (Remarks to the Author):

This is a revision of a manuscript that presents a GWAS for facial variation in the CANDELA cohort, where the authors associated genomic regions with facial features from 2D photographs. Since the last version of the manuscript, there have been many changes to clarify the methods and results described in the manuscript, and my previous comments have been addressed. I only have the following minor edits:

1. Line 131: “was superior to” – not supported by Table S1, which seems to show that the two methods perform similarly.

The reviewer is correct in that for most cases the two methods perform similarly, although for some landmarks Face++ outperforms Dlib (particularly for Face++ landmark 16, 17 (i.e. Dlib landmark 32, 36)). We have now rephrased the sentence in line 132-133 as:

“According to both metrics, the landmarks placed by Face++ were very close to the manual landmarks, and the performance of Face++ was superior to Dlib for certain landmarks.”

2. Line 143: how many individuals were removed?

We now state in line 142 the exact number of individuals with extreme values the head angles that were excluded:

“Consequently, we excluded 76 individuals with extreme values of the head angles, and included these angles as covariates in the genetic association tests.”

3. Table S3: the p-value correlation coloring is not clear in this table.

We thank the reviewer for pointing this out. Since most p-values in this table are well below the significance threshold, we have removed the coloring. Instead, we presented the numbers in scientific notation.

4. Line 110: “one region also impacts on mouse craniofacial morphology” – awkward phrasing.

We have rewritten this sentence as follows:

“For most of the novel signals identified, we find evidence of statistical replication in European, East Asian, or African GWAS data, and one mouse homologous region influences craniofacial morphology in mice.”